# Ubiquitin turnover and endocytic trafficking in yeast are regulated by Ser57 phosphorylation of ubiquitin

Sora Lee[1], Jessica M Tumolo[1], Aaron C Ehlinger[2], Kristin K Jernigan[1], Susan J Qualls-Histed[1], Pi-Chiang Hsu[3], W Hayes McDonald[2,4], Walter J Chazin[2], Jason A MacGurn[1]*

[1]Department of Cell and Developmental Biology, Vanderbilt University, Nashville, United States; [2]Department of Biochemistry, Vanderbilt University, Nashville, United States; [3]Weill Institute for Cell and Molecular Biology, Cornell University, Ithaca, United States; [4]Mass Spectrometry Research Center, Vanderbilt University, Nashville, United States

**Abstract** Despite its central role in protein degradation little is known about the molecular mechanisms that sense, maintain, and regulate steady state concentration of ubiquitin in the cell. Here, we describe a novel mechanism for regulation of ubiquitin homeostasis that is mediated by phosphorylation of ubiquitin at the Ser57 position. We find that loss of Ppz phosphatase activity leads to defects in ubiquitin homeostasis that are at least partially attributable to elevated levels of Ser57 phosphorylated ubiquitin. Phosphomimetic mutation at the Ser57 position of ubiquitin conferred increased rates of endocytic trafficking and ubiquitin turnover. These phenotypes are associated with bypass of recognition by endosome-localized deubiquitylases - including Doa4 which is critical for regulation of ubiquitin recycling. Thus, ubiquitin homeostasis is significantly impacted by the rate of ubiquitin flux through the endocytic pathway and by signaling pathways that converge on ubiquitin itself to determine whether it is recycled or degraded in the vacuole.
DOI: https://doi.org/10.7554/eLife.29176.001

*For correspondence: jason.a.macgurn@vanderbilt.edu

Competing interests: The authors declare that no competing interests exist.

## Introduction

There is mounting evidence that many human diseases – particularly diseases related to protein misfolding and aggregation such as neurodegenerative disorders - are associated with diminished function of the ubiquitin-proteasome system (UPS) and altered ubiquitin homeostasis. In general, protein aggregates found in neurodegenerative disorders (Lewy bodies in Parkinson's disease, neurofibrillary tangles in Alzheimer's disease, etc.) tend to accumulate ubiquitin, which is conjugated to aggregated proteins (*Dawson and Dawson, 2003*; *Hallengren et al., 2013*; *Upadhya and Hegde, 2007*). Dysregulation of ubiquitin homeostasis has been observed in mouse models of Huntington's disease (*Bennett et al., 2007*), spinal muscular atrophy (*Wishart et al., 2014*), amyotrophic lateral sclerosis (*Gilchrist et al., 2005*; *Hallengren et al., 2013*), and Alzheimer's disease (*Lam et al., 2000*; *Upadhya and Hegde, 2007*). Given the emerging consensus that dysregulation of ubiquitin homeostasis and UPS activity is a key feature of neurodegeneration, some have proposed restoring ubiquitin levels and UPS activity as a potential therapeutic strategy (*Chen et al., 2011*; *Chou and Deshaies, 2011*; *Deshaies, 2009*; *Lee et al., 2010*). However, very little is known about regulation of ubiquitin homeostasis in physiological conditions, during cellular aging, in response to cellular stress, or in states of disease. Thus, there is a critical need to dissect the basic mechanisms responsible for regulating ubiquitin metabolism and to identify new pathways which may be targeted to facilitate precise manipulation of ubiquitin homeostasis in the context of disease.

In yeast and mammalian cells, ubiquitin is synthesized from multiple (usually four) genes which encode ubiquitin precursor proteins consisting of either a single ubiquitin protein fused to a ribosomal protein subunit (*Finley et al., 1989*; *Ozkaynak et al., 1987*) or head-to-tail ubiquitin repeats (linear polymers) (*Finley et al., 1987*; *Ryu et al., 2008a*; *Ryu et al., 2007*; *Ryu et al., 2008b*). Encoding ubiquitin as a fusion protein to ribosomal subunits allows cells to couple ubiquitin synthesis and ribosome biogenesis in nutrient-rich conditions that support protein translation and cellular growth. In coordination with synthesis, the management of existing ubiquitin pools (including monoubiquitin, conjugated ubiquitin, and free polymers) by recycling and degradation is critical. Deubiquitylating enzymes (DUBs) that associate with protein degradation machinery (the proteasome and ESCRT machinery for soluble and membrane proteins, respectively) tend to mediate recycling of ubiquitin, and mutation of these DUBs leads to ubiquitin deficiency (*Anderson et al., 2005*; *Hallengren et al., 2013*; *Leggett et al., 2002*; *Swaminathan et al., 1999*). In particular, Doa4 – a yeast endosomal DUB that associates with the ESCRT machinery and recycles ubiquitin prior to membrane protein degradation – has a critical role in both vacuolar trafficking and management of cellular ubiquitin levels. More specifically, *doa4* mutant cells exhibit ubiquitin deficiency and vacuolar trafficking defects (*Amerik et al., 2000*; *Kimura et al., 2009*; *Swaminathan et al., 1999*), underscoring how trafficking along the endocytic route and the management of cellular ubiquitin levels are coupled. Furthermore, these findings illustrate how ubiquitin recycling is a potential point of regulation for controlling cellular ubiquitin levels yet specific mechanisms for the regulation of ubiquitin recycling and degradation remain poorly understood.

The ubiquitin code is highly complex, and modification by conjugation to ubiquitin can alter the fate of substrate proteins by promoting degradation, altering subcellular localization, or altering interactions with binding partners. The complexity of the ubiquitin code is underscored by the fact that ubiquitin can polymerize at any of seven internal lysines (or the N-terminus) leading to chains of different linkage types, each with a unique structure that can be interpreted differently, and that mixed-linkage or branched chains are also possible. More recent work has also led to a consensus that post-translational modifications of ubiquitin (other than polymerization) can alter its function (*Herhaus and Dikic, 2015*; *Zheng and Hunter, 2014*) - making the ubiquitin code as we know it even more complex than previously appreciated. For example, several groups have reported that the E3 ubiquitin ligase Parkin – mutations in which cause an autosomal recessive form of early onset Parkinson's disease – is activated by Ser65 phosphorylated ubiquitin (*Kane et al., 2014*; *Kazlauskaite et al., 2014*; *Koyano et al., 2014*; *Ordureau et al., 2014*; *Wauer et al., 2015b*). Despite the fact that the Pink1-Parkin system is not conserved in *Saccharomyces cerevisiae*, Ser65 phosphorylation of ubiquitin has been detected in yeast and recent work suggests it may play a role in the oxidative stress response (*Swaney et al., 2015*). These findings suggest that Ser65 phosphoregulation of ubiquitin is highly conserved across evolution and likely has functional significance beyond Pink1-Parkin-mediated mitophagy. Importantly, several other studies have reported evidence of phosphorylation at other sites on ubiquitin in yeast (*Peng et al., 2003*) and mammalian (*Villén et al., 2007*) cells. While biochemical analysis has revealed that different phosphorylation events on ubiquitin impact the *in vitro* activities of specific E2-conjugating enzymes, E3 ubiquitin ligases, and deubiquitylases (*Huguenin-Dezot et al., 2016*; *Wauer et al., 2015b*) the actual functional significance of these modifications remain to be elucidated.

Here, we report that a pair of highly similar yeast phosphatases – Ppz1 and Ppz2 – are required for proper management of ubiquitin homeostasis in yeast. We show that *ppz* mutants exhibit elevated levels of Ser57 phosphorylated ubiquitin – suggesting that Ser57 phosphorylation of ubiquitin may be linked to the regulation of ubiquitin homeostasis. We show that phosphomimetic mutations at the Ser57 position confer increased rate of ubiquitin degradation and gain-of-function endocytic trafficking phenotypes. Furthermore, we present evidence that these phenotypes are associated with resistance to removal by deubiquitylases on yeast endosomes, and that this pathway plays a significant role in the regulation of ubiquitin metabolism. Based on these findings, we propose Ser57 phosphorylation of ubiquitin as a potential mechanism for deciding whether ubiquitin is recycled or degraded during multi-vesicular body (MVB) sorting on endosomes, thus contributing to the global regulation of ubiquitin levels in the cell.

## Results

### Ppz phosphatases regulate ubiquitin phosphorylation and homeostasis

As part of an ongoing effort to elucidate signaling pathways that regulate endocytic trafficking in yeast, we recently became interested in a pair of highly similar (57% identical) protein phosphatases in *Saccharomyces cerevisiae* called Ppz1 and Ppz2 given their reported role in the regulation of ion transporter function (*Ruiz et al., 2004*; *Ruiz et al., 2006*; *Yenush et al., 2005*). To explore their potential role in endocytic trafficking and to identify Ppz substrates, we performed a SILAC-based quantitative analysis of the phosphoproteomes of wild-type and Δ*ppz1*Δ*ppz2* (or *ppz*) mutant cells (*Albuquerque et al., 2008*; *MacGurn et al., 2011*). We identified several phosphorylation events elevated in *ppz* mutants, including an ~3-fold increase in the phosphorylation of ubiquitin at the Ser57 position (*Figure 1—figure supplement 1* and *Figure 1—source data 1*). Phosphorylation of ubiquitin at the Ser57 position has been reported previously (*Peng et al., 2003*), although this serine is not required for the essential function of ubiquitin (*Sloper-Mould et al., 2001*). Ubiquitin phosphorylated at the Ser57 position was also detected by a phospho-specific antibody capable of detecting Ser57 phosphorylated mono- and di-ubiquitin in yeast cell lysates (*Figure 1—figure supplement 2*). Consistent with the quantitative phosphoproteomic analysis, elevated levels of Ser57 phosphorylated ubiquitin were detected with the phospho-specific antibody in *ppz* mutant cell lysates (*Figure 1A*). Taken together, these data indicate that Ser57 phosphorylation of ubiquitin is elevated in *ppz* mutant cells.

Since the regulatory function of Ser57 phosphorylation of ubiquitin has not been described, we considered the possibility that Ppz phosphatases might regulate some aspect of ubiquitin biology via the regulation of Ser57 phosphorylation. Immunoblotting analysis revealed that *ppz* mutants exhibit mono-ubiquitin and total ubiquitin deficiencies (*Figure 1B–D* and *Figure 1—figure supplement 3–4*). Importantly, complementation analysis revealed that addition of either *PPZ1* or *PPZ2* could complement the ubiquitin deficiency observed in *ppz* mutant cells (*Figure 1B–C* and *Figure 1—figure supplement 4*). Furthermore, an R451L catalytic dead mutant of Ppz1 was unable to complement the ubiquitin deficiency (*Figure 1B–C*), indicating that Ppz1 phosphatase activity is critical for ubiquitin management. We next considered that the ubiquitin deficiency observed in *ppz* mutant cells might be related to the regulation of ubiquitin metabolism. To explore this possibility, we performed cycloheximide chase experiments and measured ubiquitin turnover in wild-type and *ppz* mutant cells, which revealed that ubiquitin degradation occurs faster in *ppz* mutants (*Figure 1E–F*). To our knowledge, this is the first example of a signaling protein required for proper regulation of ubiquitin homeostasis.

In the course of these experiments, we observed several *ppz* mutant phenotypes including resistance to LiCl and $MnCl_2$ and hypersensitivity to KCl and caffeine (*Figure 1G* and *Figure 1—figure supplement 5*). Strikingly, we found that ubiquitin overexpression (driven by the ADH1 promoter) in *ppz* mutant cells suppressed these phenotypes (*Figure 1G*), indicating that these phenotypes are caused by ubiquitin deficiency. Importantly, *doa4* mutants – which are also ubiquitin deficient – similarly exhibit phenotypes that can be rescued by ubiquitin supplementation (*Amerik et al., 2000*; *Swaminathan et al., 1999*).

Based on our findings that *ppz* mutants exhibit defects in ubiquitin homeostasis and elevated phosphorylation of ubiquitin at the Ser57 position, we hypothesized that elevated Ser57 phosphorylation might contribute to the ubiquitin deficiency observed in *ppz* mutants. To test this, we analyzed total ubiquitin levels in wild-type and *ppz* mutant cell lysates (SUB280 strain background) expressing either wild-type or Ser57Ala mutant ubiquitin. This analysis revealed that expression of Ser57Ala mutant ubiquitin could partially suppress the ubiquitin deficiency observed in *ppz* mutant cells (*Figure 2A and B*). To test this further, we expressed FLAG-ubiquitin in wild-type and *ppz* mutant cells (SUB280 strain background) and performed a cycloheximide chase to measure the rate of ubiquitin degradation (*Figure 2C*). Consistent with prior results (*Figure 1E and F*), this analysis revealed that ubiquitin is degraded faster in *ppz* mutant cells (*Figure 2D*) but this increased ubiquitin turnover can be suppressed by expression of Ser57Ala mutant ubiquitin (*Figure 2E*). We also found that some of the *ppz* mutant phenotypes driven by ubiquitin deficiency (*Figure 1G*) were partially suppressed by expression of Ser57Ala mutant ubiquitin. Specifically, we observed that expression of Ser57Ala mutant ubiquitin in *ppz* mutant cells partially suppressed $MnCl_2$ resistance, KCl sensitivity, and caffeine sensitivity, although it had no affect on the LiCl resistance phenotype (*Figure 2F*).

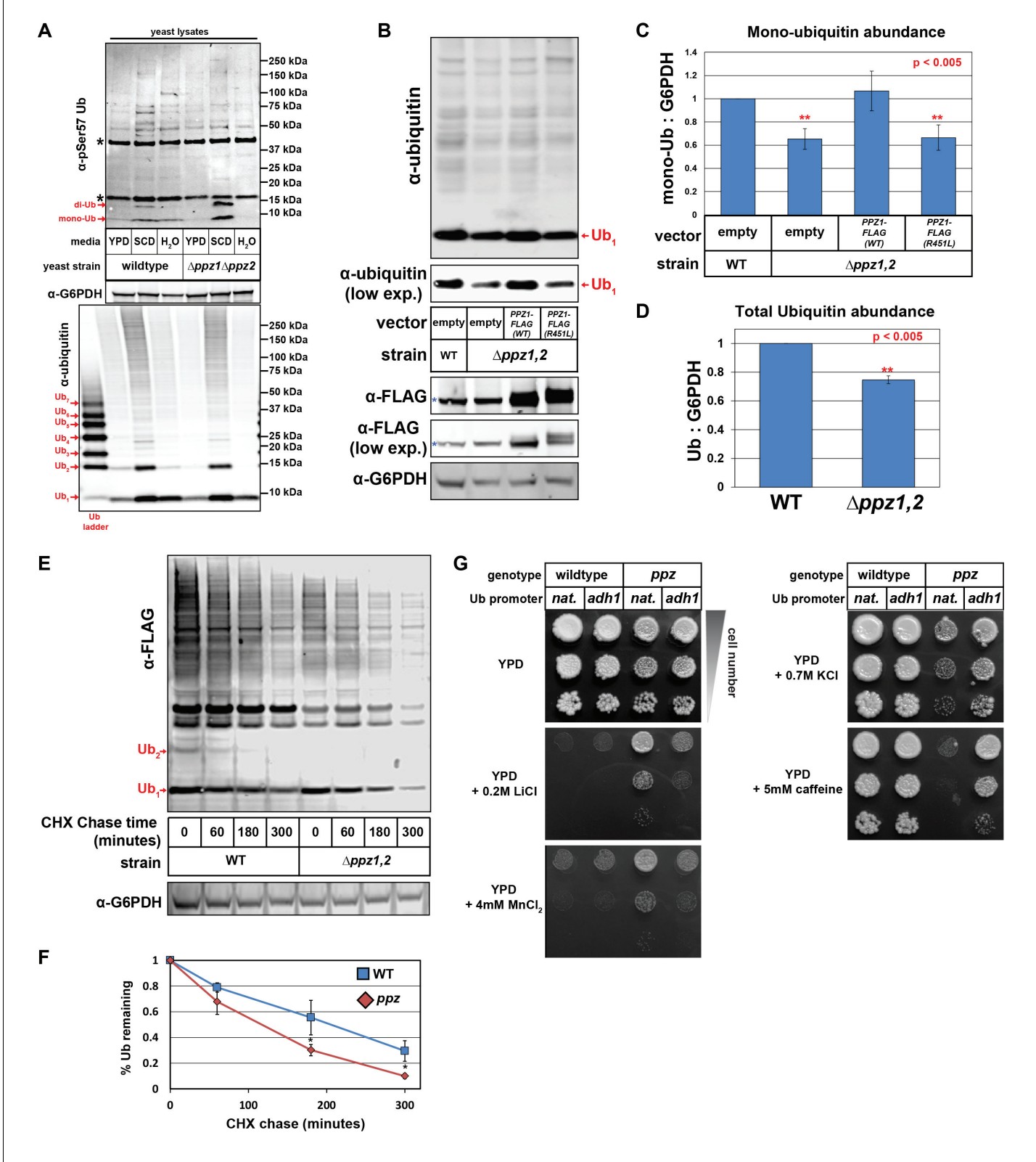

**Figure 1.** Ppz phosphatase activity is required for proper management of cellular ubiquitin. (**A**) Immunoblot analysis using an α-phospho-Ser57 specific antibody to detect Ser57 phosphorylation of ubiquitin in yeast lysates. Yeast lysates from a SUB280 background, comparing wild-type cells (left three lanes) and *ppz* mutant cells (right three lanes) grown to mid log phase in YPD ('YPD'), or shifted from mid log growth in YPD to minimal complete media for 6 hr ('SCD') or shifted to growth in water overnight ('H₂O'). Samples were resolved by SDS-PAGE and immunoblotted using antibodies that

*Figure 1 continued on next page*

*Figure 1 continued*

recognize total ubiquitin (bottom panel), G6PDH (middle panel), or phospho-Ser57 ubiquitin (top panel). For the total ubiquitin blot (bottom panel), a ubiquitin ladder was included as a standard for different unconjugated poly-ubiquitin species (red arrows). For the α–phospho-Ser57 ubiquitin blot (top panel), red arrows indicate species that are specific to Ser57 on ubiquitin and asterisks (*) indicate non-specific bands detected by the antibody, as illustrated in *Figure 1—figure supplement 2*. (B) The indicated yeast cells (SEY6210 background) containing either empty vector or *PPZ1-FLAG* vectors were analyzed for total cellular ubiquitin levels by immunoblot analysis. The blue asterisk indicates a background band detectable by FLAG antibody that is a MW similar to Ppz1-FLAG. (C) Quantification of mono-ubiquitin abundance in yeast cell lysates (SEY6210 background) from multiple biological replicates of the immunoblot shown in (B) (n = 5). (D) Total ubiquitin abundance was quantified in wild-type and *ppz* mutant cells expressing endogenously FLAG-tagged ubiquitin. Total cell lysates were analyzed by quantitative analysis of slot blots shown in *Figure 1—figure supplement 3*. (E) The indicated yeast cells were grown to mid-log phase and cell lysates were analyzed for total ubiquitin levels at the indicated time points following a cycloheximide (CHX) chase. (F) The results for (F) were quantified over multiple experiments (n = 3). (G) Analysis of yeast growth in the indicated conditions. In this experiment, the yeast ubiquitin gene was expressed from either native (pRPS31) or overexpression (pADH1) promoter was shuffled into the SUB280 strain background. Indicated yeast were plated in 10-fold serial dilutions on indicated plates. In all panels, double asterisk (**) indicates p<0.005 and single asterisk (*) indicates p<0.05.

DOI: https://doi.org/10.7554/eLife.29176.002

The following source data and figure supplements are available for figure 1:

**Source data 1.** Results from SILAC-based quantitative comparison of the yeast phosphoproteome from wild-type (heavy) and Δ*ppz1*Δ*ppz2* (light) cells.
DOI: https://doi.org/10.7554/eLife.29176.008

**Source data 2.** This spreadsheet contains the quantification and statistical analysis for mono-ubiquitin levels (*Figure 1C*), total ubiquitin levels (*Figure 1D*), and for ubiquitin degradation in a cycloheximide chase experiment (*Figure 1F*).
DOI: https://doi.org/10.7554/eLife.29176.009

**Figure supplement 1.** A SILAC-based quantitative comparison of the phosphoproteome in wildtype (heavy) and Δ*ppz1*Δ*ppz2* (light) cells.
DOI: https://doi.org/10.7554/eLife.29176.003

**Figure supplement 2.** Analysis of antibodies recognizing pSer57 ubiquitin.
DOI: https://doi.org/10.7554/eLife.29176.004

**Figure supplement 3.** Quantification of cellular ubiquitin levels by analysis of slot blots.
DOI: https://doi.org/10.7554/eLife.29176.005

**Figure supplement 4.** Characterization of ubiquitin levels and distribution in wild-type and Δ*ppz1*Δ*ppz2* cells.
DOI: https://doi.org/10.7554/eLife.29176.006

**Figure supplement 5.** Catalytic activity of the Ppz1 phosphatase is required for phenotype complementation.
DOI: https://doi.org/10.7554/eLife.29176.007

Taken together, these results indicate that elevated phosphorylation of ubiquitin at the Ser57 position contributes partially to the ubiquitin deficiency observed in *ppz* mutant cells.

## Ubiquitin turnover is regulated by the Ser57 position of ubiquitin

Given our results linking the regulation of ubiquitin metabolism to Ser57 phosphorylation in *ppz* mutant cells, we hypothesized that Ser57 phosphorylation may play a role in the regulation of cellular ubiquitin metabolism. To test this, we generated yeast strains where a single endogenous ubiquitin locus (*RPS31*) has been modified by homologous recombination to express N-terminally FLAG-tagged ubiquitin that is either wildtype, Ser57Ala (phosphorylation resistant), or Ser57Asp (phosphomimetic) and measured ubiquitin half-life in these strains following a cycloheximide chase. We found that Ser57Ala ubiquitin turned over more slowly than wild-type ubiquitin, while Ser57Asp ubiquitin turned over faster than wild-type ubiquitin (*Figure 3—figure supplement 1*). To explore this further, we measured ubiquitin half-life using a galactose induction/glucose repression system and found that Ser57Asp ubiquitin exhibits a significantly shorter half-life than wildtype (*Figure 3A–D*) while Ser57Ala ubiquitin exhibited a half-life equivalent to wild-type ubiquitin (*Figure 3C*). These data suggest that Ser57 phosphorylation may regulate ubiquitin turnover in the cell since mutations that prevent or mimic phosphorylation modulate ubiquitin turnover.

While the half-life analysis of Ser57 phosphomimetic ubiquitin suggests that phosphorylation may be sufficient to promote ubiquitin degradation, the corresponding analysis of the Ser57Ala mutant indicates that Ser57 phosphorylation may be required for ubiquitin turnover in some conditions (e.g. the cycloheximide chase in *Figure 3—figure supplement 1*) but not in others (e.g. the galactose induction/glucose repression chase experiment in *Figure 3C*). One possible explanation for these findings is that Ser57 phosphorylation of ubiquitin occurs transiently, at low stoichiometry or in a highly localized manner. In an attempt to quantify the stoichiometry of Ser57 phosphorylation, we

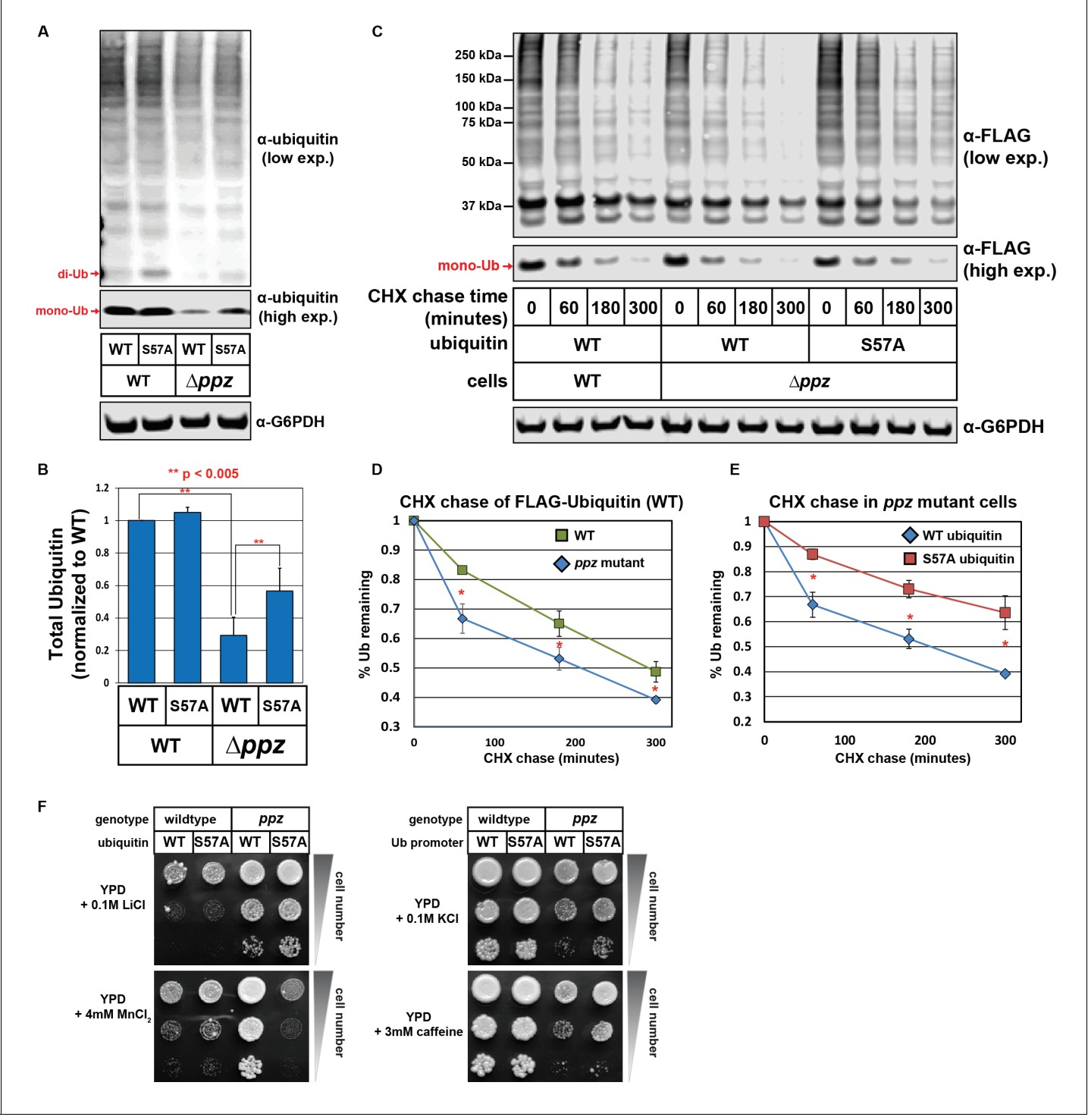

**Figure 2.** Ser57 phosphorylation of ubiquitin contributes to *ppz* mutant phenotypes. (**A**) Quantitative immunoblot analysis was performed on lysates from wild-type or *ppz* mutant yeast cells (SUB280 background) expressing either wild-type ubiquitin or Ser57Ala ubiquitin. (**B**) The results for (**A**) were quantified over multiple experiments (n = 3). (**C**) Quantitative immunoblot analysis was performed on lysates from wild-type or *ppz* mutant yeast cells (SUB280 background) expressing either wild-type ubiquitin or Ser57Ala FLAG-ubiquitin (driven by the TEF1 promoter) following addition of cycloheximide (CHX). (**D and E**) The results for (**C**) were quantified over multiple experiments (n = 3). (**F**) Wild-type and *ppz* mutant yeast strains (SUB280 background) expressing only wild-type or Ser57Ala ubiquitin were plated in 10-fold serial dilution on the indicated media. Plates were imaged after three days of growth at 26°C. In all panels, double asterisk (**) indicates p<0.005 and single asterisk (*) indicates p<0.05.

DOI: https://doi.org/10.7554/eLife.29176.010

*Figure 2 continued on next page*

*Figure 2 continued*

The following source data is available for figure 2:

**Source data 1.** This spreadsheet contains the quantification and statistical analysis for total ubiquitin levels (*Figure 2B*) and for ubiquitin degradation in a cycloheximide chase experiment (*Figure 2D–E*).
DOI: https://doi.org/10.7554/eLife.29176.011

performed multiple reaction monitoring (MRM) mass spectrometry analysis on ubiquitin affinity purified from *ppz* mutant cell lysates and spiked with isotopically labeled (heavy) standard peptides corresponding to unmodified and phosphorylated Ser57. Ser57 phosphorylation was detected in these samples but was present at a level below that required for reliable quantification – although it is clear based on internal heavy standard measurements that Ser57 phosphorylation occurs on less than 0.05% of total ubiquitin in *ppz* mutant cell lysates (*Figure 3—figure supplement 2*). These data suggest that under normal growth conditions Ser57 phosphorylation occurs at very low stoichiometry and may function to regulate the turnover of a limited and/or highly localized portion of the ubiquitin pool in physiological conditions. Although the rate of ubiquitin turnover is increased in *ppz* mutants and for Ser57Asp phosphomimetic ubiquitin, the low stoichiometry of Ser57 phosphorylation (even in *ppz* mutants) limits our ability to determine the relative contribution of this modification to the overall regulation of ubiquitin homeostasis.

Given our observation that Ser57 phosphomimetic ubiquitin increased the rate of turnover, we also considered the possibility that this mutation causes loss of ubiquitin structure. To test this, we used NMR to assay the tertiary structure of Ser57Asp ubiquitin relative to wildtype. $^{15}$N-$^{1}$H heteronuclear single quantum coherence (HSQC) experiments are highly sensitive to changes in the electronic environment of backbone amides, and can readily detect even subtle changes in structure. Overlay of the spectra of wildtype and mutant proteins reveals differences in the position or intensity of signals. *Figure 3E* shows that the Ser57Asp mutation causes only a small number of perturbations. Cross-referencing to the chemical shift assignments (BMRB entry 17769) and structure (PDB entry 1UBQ) of ubiquitin allowed us to localize these effects to a surface immediately adjacent to the mutation site (*Figure 3F*). The small number of perturbations observed here is fully consistent with previous studies of mutations at surface exposed sites resulting in only minor changes in the local environment. Because the Ser57Asp mutant ubiquitin retains a highly similar structure to wildtype ubiquitin (*Figure 3F*), we can conclude that the decreased half-life observed for Ser57Asp ubiquitin *in vivo* (*Figure 3A–D* and *Figure 3—figure supplement 1*) is not due to disruption of ubiquitin structure.

## Ser57 phosphomimetic ubiquitin accelerates endocytic trafficking

We decided to explore further how ubiquitin phosphorylation at Ser57 could alter its metabolism in the cell. We hypothesized that increased ubiquitin metabolism could result from increased degradation of ubiquitin by either the proteasome or the vacuole. The later possibility could occur via increased flux of ubiquitin through the endocytic pathway – which we decided to test by examining growth in the presence of canavanine. Canvanine is a toxic arginine analog that only enters yeast cells via the arginine transporter Can1, and mutations that disrupt or hyperactivate Can1 endocytic trafficking confer canavanine hypersensitivity or resistance, respectively (*Lin et al., 2008*). Surprisingly, we found that yeast cells expressing Ser57Asp phosphomimetic ubiquitin conferred resistance to canavanine (*Figure 4A and B*) – indicative of a gain-of-function endocytic trafficking phenotype (*Lin et al., 2008*; *MacGurn et al., 2011*). This was observed when ubiquitin was expressed exogenously from a plasmid (in addition to endogenous ubiquitin) (*Figure 4A*) or when it was expressed as the sole source of ubiquitin from a single locus (*Figure 4B*). Importantly, we found that mutations which prevent ubiquitin conjugation (G75V, G76V and ΔG75, ΔG76) suppressed the Ser57Asp canavanine resistance phenotype (*Figure 4—figure supplement 1*), indicating that conjugation is required for enhanced endocytic trafficking. Similarly, we tested if K63- or K48-linked polyubiquitin is required for the endocytic gain-of-function associated with expression of Ser57Asp phosphomimetic ubiquitin and found that both linkage types are required for full canavanine resistance (*Figure 4—figure supplement 2*). Taken together, these experiments demonstrate that the canavanine

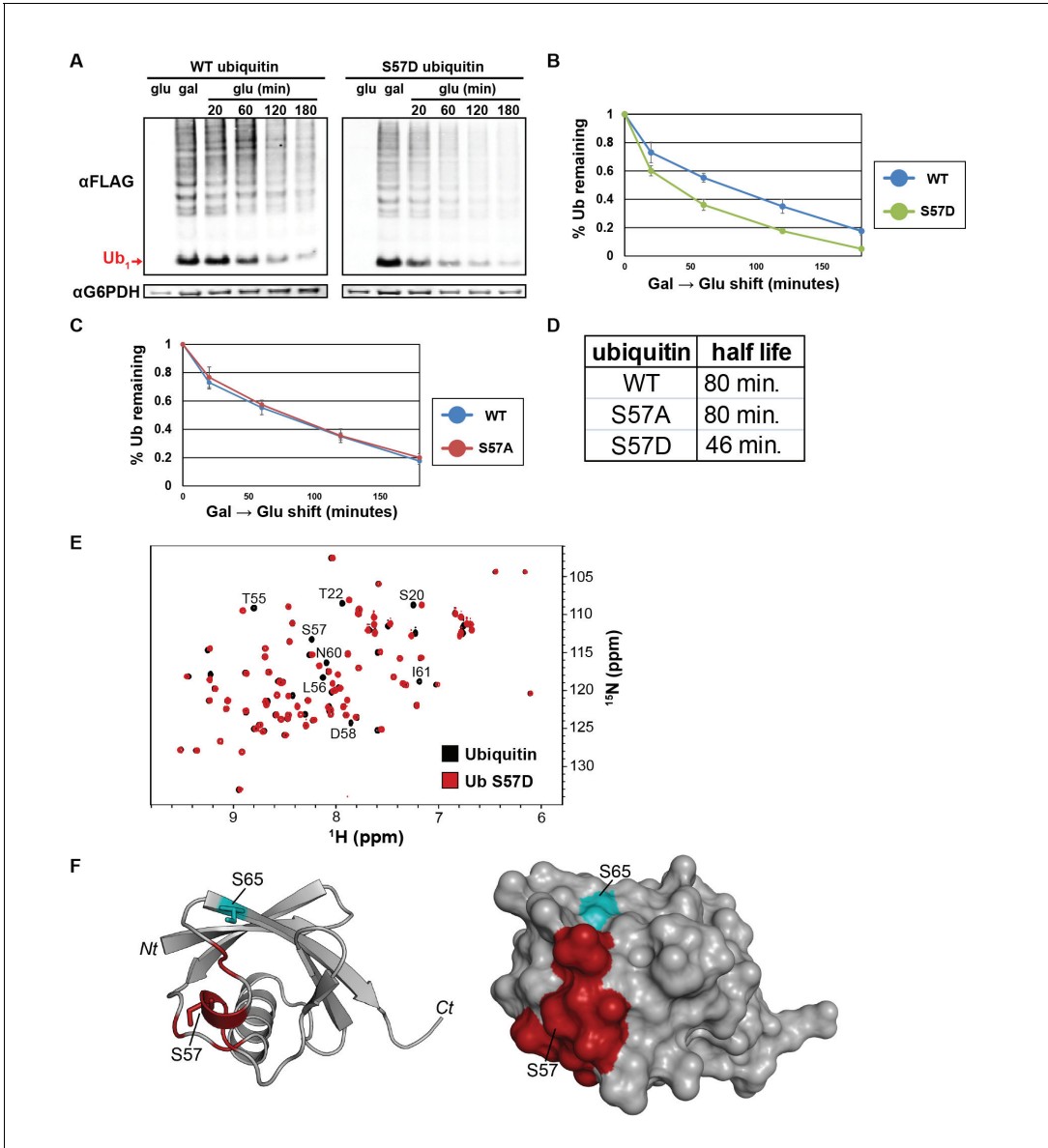

**Figure 3.** The Ser57 position of ubiquitin is a critical determinant of ubiquitin metabolism. (**A**) SDS-PAGE immunoblot analysis of lysates from cells expressing FLAG-tagged ubiquitin from the *GAL10* galactose-inducible promoter. Yeast strains were grown to mid-log phase in glucose media, induced to express FLAG-ubiquitin by shifting to galactose-containing media overnight, and then shifted back to glucose-containing media to repress further transcription of FLAG-ubiquitin. The top panel shows α–FLAG immunoblots and the bottom panel shows immunoblots of G6PDH, a loading control. Mono-ubiquitin is indicated by the red arrow. (**B and C**) Quantitation of ubiquitin degradation for wildtype, S57D (**B**) and S57A (**C**) ubiquitin was averaged (n = 4) with error bars indicating standard deviation. (**D**) Ubiquitin half-life was estimated based on trendline regression analysis. (**E**) Overlay of $^{15}N$-$^{1}H$ HSQC spectra generated for wild-type ubiquitin (black) and Ser57Asp phosphomimetic ubiquitin (red). Residues that are significantly perturbed in the phosphomimetic are labeled. (**F**) Residues perturbed in the Ser57Asp phosphomimetic (from *Figure 3E*) were mapped onto the structure of ubiquitin (PDB entry 1UBQ) in red. Key phosphorylation sites at Ser57 and Ser65 (cyan) are labeled.

DOI: https://doi.org/10.7554/eLife.29176.012

The following figure supplements are available for figure 3:

**Figure supplement 1.** Analysis of ubiquitin half-life by SDS-PAGE immunoblot analysis of yeast lysates following a cycloheximide (CHX) chase (top panel).

DOI: https://doi.org/10.7554/eLife.29176.013

**Figure supplement 2.** Analysis of stoichiometry of Ser57 phosphorylation of ubiquitin.

DOI: https://doi.org/10.7554/eLife.29176.014

resistance phenotype caused by Ser57Asp ubiquitin requires both substrate conjugation and formation of polyubiquitin linkages.

To further explore endocytic trafficking in cells expressing Ser57Asp phosphomimetic ubiquitin, we examined the methionine-induced trafficking and degradation of the methionine transporter Mup1. Although the endocytosis and vacuolar degradation of Mup1 is already extremely rapid following stimulation of the cells by addition of methionine (*Lin et al., 2008*; *MacGurn et al., 2011*), we found that expression of Ser57Asp phosphomimetic ubiquitin resulted in accelerated Mup1 turnover in response to methionine addition (*Figure 4—figure supplement 3*). Furthermore, we found that lower concentrations of methionine had minimal effect on Mup1 turnover in the presence of wildtype ubiquitin but triggered rapid and efficient turnover of Mup1 in the presence of Ser57Asp phosphomimetic ubiquitin (*Figure 4C and D*). These results indicate that Ser57Asp ubiquitin sensitizes Mup1 to endocytic trafficking and vacuolar degradation. Although these findings suggest that Ser57 phosphorylation of ubiquitin would be sufficient to promote the trafficking of cargo to the vacuole for degradation, the low stoichiometry observed for this modification (*Figure 3—figure supplement 2*) limits our ability to determine the relative role this modification plays in the overall regulation of endocytic trafficking.

## Ser57 phosphomimetic ubiquitin bypasses an artificial DUB checkpoint

We hypothesized that the gain-of-function endocytic trafficking phenotypes observed in the presence of Ser57Asp phosphomimetic ubiquitin could be due to (i) increased rate of ubiquitin conjugation, (ii) altered affinity for ubiquitin binding domains (UBDs) that mediate cargo recognition and sorting along the endocytic pathway, or (iii) altered recognition by deubiquitylating enzymes (DUBs) that functions as checkpoints along the endocytic route to promote recycling of both cargo and ubiquitin. To test if Ser57 phosphorylation of ubiquitin alters the rate of ubiquitin conjugation, we reconstituted *in vitro* the E3 ubiquitin ligase activity of Rsp5 (*Kim and Huibregtse, 2009*), a major regulator of endocytic trafficking in yeast (*Lauwers et al., 2010*; *MacGurn et al., 2012*), using purified recombinant proteins. This analysis revealed that Ser57Asp phosphomimetic mutations do not affect Rsp5 E3 ubiquitin ligase activity *in vitro* (*Figure 5—figure supplement 1*). We next considered if phosphorylation affects interaction with UBDs along the endocytic route. We found that Ser57Asp phosphomimetic ubiquitin did not exhibit altered binding to recombinant purified ESCRT-I or ESCRT-II complexes (*Figure 5—figure supplement 2*), each of which contain multiple ubiquitin binding elements (*Dikic et al., 2009*; *Shields et al., 2009*). Finally, we considered the possibility that Ser57 phosphorylation of ubiquitin alters recognition by deubiquitylases along the endocytic trafficking route. To test this, we took advantage of the previously published observation that fusing deubiquitylase domains to ESCRT-0 prevents cargo sorting on endosomes and thus prevents delivery of cargo to the lumen of the vacuole (*Stringer and Piper, 2011*). Normally, endocytic cargos like Mup1 are trafficked along the endocytic route, from the plasma membrane to the lumen of the vacuole, in response to the addition of methionine to the media (*Figure 5A*) (*Lin et al., 2008*). However, fusing the UL36 deubiquitylase domain to Hse1 (part of ESCRT-0) prevents delivery of Mup1 to the vacuole lumen, instead causing it to accumulate on the limiting membrane of the vacuole in response to methionine stimulation (*Figure 5B*, top panel). Surprisingly, we found that expression of Ser57Asp ubiquitin facilitated bypass of this artificial DUB checkpoint, allowing Mup1 to traffic to the lumen of the vacuole (*Figure 5B*, bottom panel). We also observed that fusion of Hse1 to catalytically active (but not catalytically dead) UL36 conferred hypersensitivity to canavanine – consistent with a trafficking block in these strains – which could be suppressed by the expression of Ser57Asp (but not wildtype) ubiquitin (*Figure 5C*). Importantly, suppression of the canavanine hypersensitive phenotype by Ser57Asp ubiquitin required components of ESCRT-I (Vps23) and ESCRT-III (Snf7), but not components of the autophagy pathway (Atg5, Atg8) (*Figure 5D* and *Figure 5—figure supplement 3*), indicating that Ser57Asp ubiquitin restores cargo delivery to the vacuole lumen through the ESCRT pathway and not a parallel pathway. To further test the ability of Ser57Asp ubiquitin to bypass DUB recognition, we fused the arginine transporter Can1 to UL36 and found that this also confers canavanine hypersensitivity that can be partially suppressed by expression of Ser57Asp ubiquitin (*Figure 5—figure supplement 4*). All these results indicate that in a cellular context, and more specifically on endosomal membranes, Ser57Asp ubiquitin bypasses DUB recognition in order to promote cargo trafficking to the vacuole lumen. Thus, given the low observed stoichiometry of Ser57 phosphorylation (*Figure 3—figure supplement 2*), it is possible that phosphorylation and dephosphorylation of

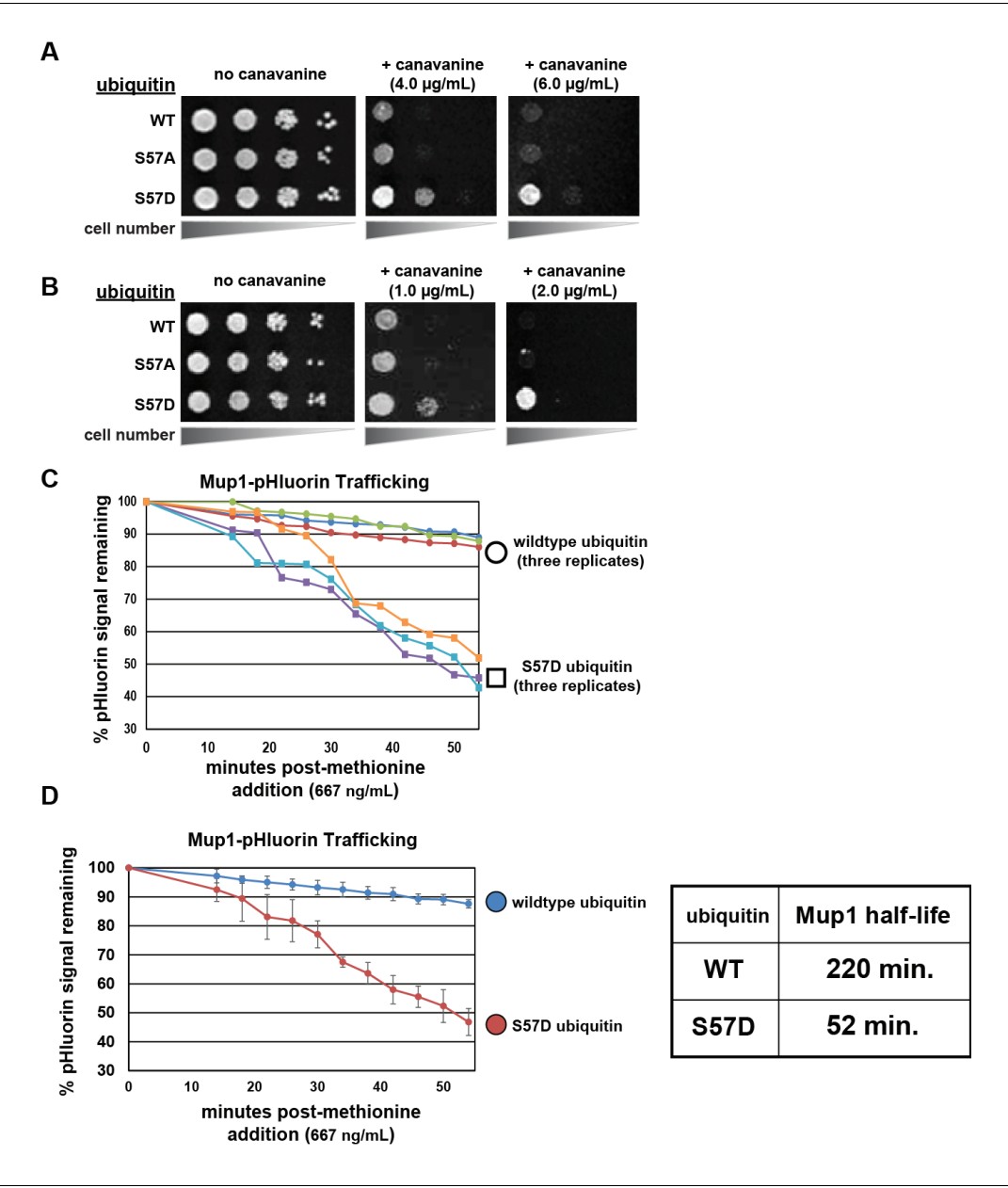

**Figure 4.** Ser57 phosphomimetic mutations in ubiquitin accelerate endocytic trafficking. (**A**) Analysis of yeast growth in the presence of canavanine. In this experiment, the indicated ubiquitin variants (wildtype, Ser57Ala, or Ser57Asp) were expressed exogenously from the pRS416 plasmid under the control of the *ADH1* promoter in the SEY6210 strain background. Canavanine hypersensitivity is indicative of an endocytic defect, while canavanine resistance is indicative of hyper-active endocytic trafficking. (**B**) Analysis of yeast growth in the presence of canavanine. Here, the indicated ubiquitin variants (wildtype, Ser57Ala, or Ser57Asp) were expressed as the sole source of ubiquitin from the native *RPS31* promoter in the SUB280 strain background. (**C**) Flow cytometry analysis of yeast cells expressing Mup1-pHluorin grown in the absence of methionine and then stimulated by addition of a low concentration of methionine (667ng/mL). Fluorescence of pHluorin is pH-sensitive and lost during endocytic trafficking when the cargo encounters an acidic environment. Triplicate experiments are shown for cells expressing wildtype (circles) or S57D phosphomimetic (squares) ubiquitin. (**D**) Data generated in (**C**) were averaged (n = 3, error bars indicate standard deviation) and linear regression was used to generate trend lines and estimate the half-life of Mup1-pHluorin in the context of wildtype or Ser57Asp ubiquitin, as indicated in the table (right).

DOI: https://doi.org/10.7554/eLife.29176.015

The following figure supplements are available for figure 4:

*Figure 4 continued on next page*

*Figure 4 continued*

**Figure supplement 1.** Hyper-active endocytic trafficking in the presence of Ser57Asp phosphomimetic ubiquitin requires conjugation.
DOI: https://doi.org/10.7554/eLife.29176.016
**Figure supplement 2.** Hyper-active endocytic trafficking in the presence of Ser57Asp phosphomimetic ubiquitin requires K63-linked poly-ubiquitination.
DOI: https://doi.org/10.7554/eLife.29176.017
**Figure supplement 3.** Expression of Ser57Asp phosphomimetic ubiquitin accelerates methionine-stimulated endocytic trafficking of Mup1.
DOI: https://doi.org/10.7554/eLife.29176.018

limited pools of ubiquitin localized to the endosome may impact rates of ubiquitin recycling and turnover.

Our data indicate that phosphorylation of ubiquitin may increase both the rate of endocytic trafficking and the turnover of ubiquitin, and we hypothesized that bypass of endocytic DUB checkpoints might explain both these findings if normal ubiquitin degradation is dependent on endocytic trafficking and vacuolar degradation. To test this, we measured the rate of ubiquitin degradation in strains expressing catalytically active or catalytically dead UL36 fused to Hse1 and found that installing the artificial DUB checkpoint significantly decreases the rate of ubiquitin degradation (*Figure 5E, F and G*). Importantly, expression of Ser57Asp ubiquitin increases the rate of ubiquitin degradation despite the artificial DUB checkpoint (*Figure 5H, I and J*). These results illustrate how endosomal DUB bypass by Ser57 phosphorylated ubiquitin can regulate both endocytic trafficking and the rate of ubiquitin degradation.

We next explored the possibility that the accelerated trafficking observed in the presence of Ser57Asp ubiquitin might be a general feature associated with expression of DUB-resistant ubiquitin. To test this, we analyzed the effect of expressing a known DUB-resistant ubiquitin mutant (Leu73Pro) (*Békés et al., 2013*) in yeast. Importantly, in contrast to Ser57Asp ubiquitin, the Leu73Pro mutant when expressed as the sole source of ubiquitin could not support viability (*Figure 5—figure supplement 5A*) – indicating that Ser57Asp mutant ubiquitin is not broadly DUB resistant. Furthermore, in contrast to Ser57Asp, exogenous expression of Leu73Pro mutant ubiquitin conferred hypersensitivity to canavanine and delayed Mup1 trafficking (*Figure 5—figure supplement 5*) – indicating that broad defects in deubiquitylase activity confer endocytic trafficking defects that are distinct from the endocytic gain-of-function effects observed in the presence of Ser57Asp ubiquitin. Thus, our results suggest that Ser57Asp does not confer broad inhibition of cellular DUB activities but rather may affect recognition by specific DUBs along the endocytic route.

## Ser57Asp and pSer57 ubiquitin are less susceptible to cleavage by Doa4

We next decided to examine how Ser57 phosphorylation of ubiquitin affects recognition by Doa4, which is thought to be the main deubiquitylase responsible for recycling ubiquitin prior to MVB sorting on the endosome (*Amerik et al., 2000*; *Dupré and Haguenauer-Tsapis, 2001*; *Swaminathan et al., 1999*). To test this, we reconstituted Doa4 activity *in vitro* using ubiquitin-conjugated Art1 (generated from *in vitro* conjugation reactions, as in *Figure 5—figure supplement 1*) as a model substrate. As previously reported, significant Doa4 activity *in vitro* was only detected in the presence of Bro1 (*Figure 6A*), a yeast homolog of the human ALIX protein which is thought to bind and activate Doa4 on endosomes (*Luhtala and Odorizzi, 2004b*; *Pashkova et al., 2013*; *Richter et al., 2007*). Importantly, we observed that Doa4 activity was significantly reduced toward Ser57Asp phosphomimetic ubiquitin (*Figure 6A and B*). We similarly tested Doa4 activity toward synthetic unmodified and Ser57 phosphorylated ubiquitin and found that Ser57 phosphorylation decreased Doa4 activity (*Figure 6C, D and E*). These findings are consistent with the hypothesis that Ser57 phosphorylation prevents ubiquitin recycling along the endocytic route by interfering with Doa4 recognition.

Doa4 has been reported to play an important role in ubiquitin homeostasis at least in part due to its ability to recycle ubiquitin transiting through the MVB pathway (*Kimura and Tanaka, 2010*; *Kimura et al., 2009*; *Swaminathan et al., 1999*), which is consistent with our finding that Δ*doa4*

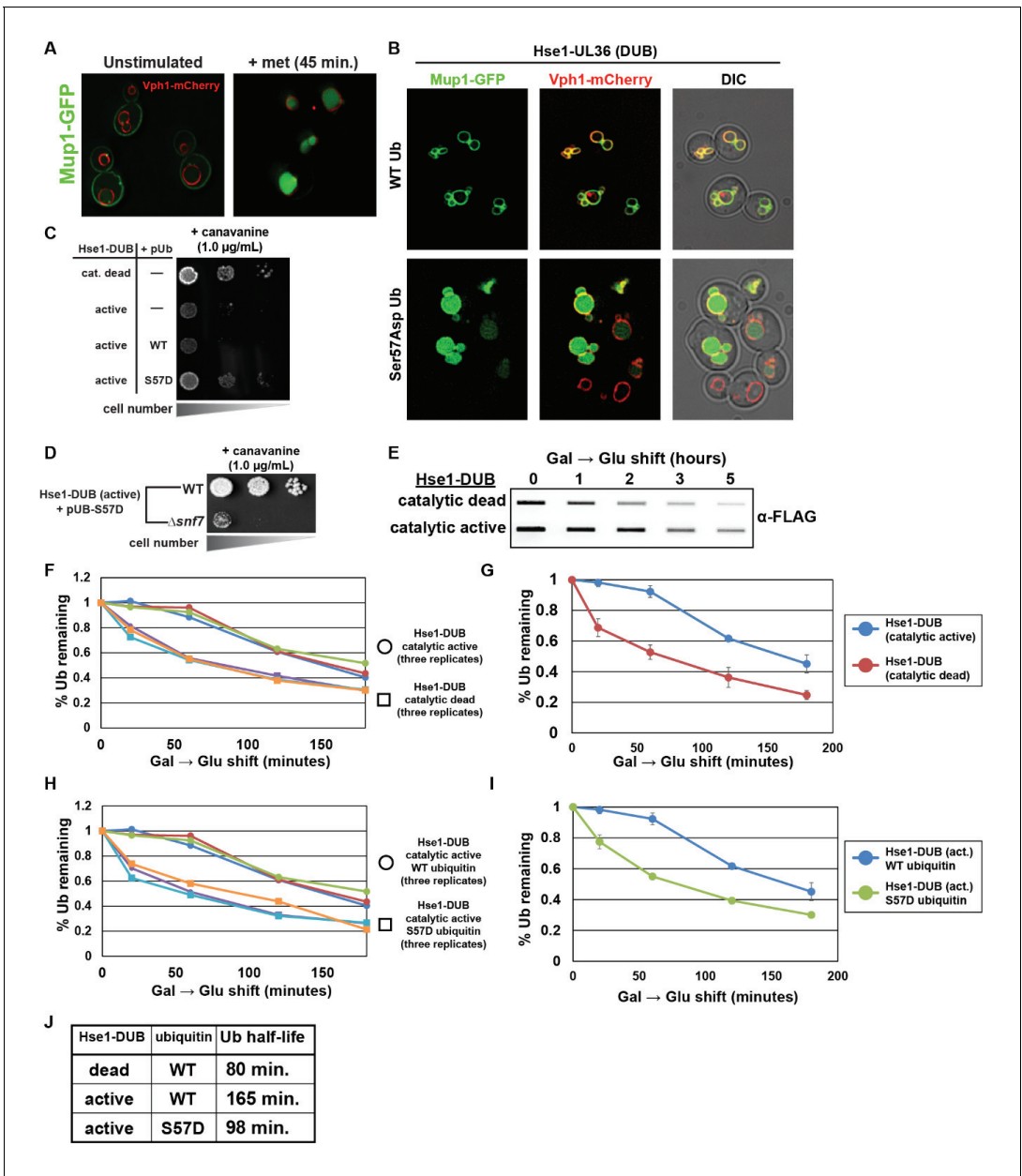

**Figure 5.** Ser57Asp phosphomimetic ubiquitin bypasses an artificial DUB checkpoint. (**A**) Analysis of Mup1-GFP (green) localization in wildtype yeast cells (strain SEY6210) grown in minimal media lacking methionine (left) or stimulated with methionine for 45 min (right). Cells are stably expressing Vph1-mCherry (red), a marker of the vacuolar membrane. (**B**) Mup1-GFP localization in yeast cells (strain SEY6210) where the endogenous Hse1 locus has been fused to the UL36 deubiquitylating enzyme, encoding an Hse1-UL36 fusion protein. This creates an artificial DUB checkpoint at ESCRT-0. The top panel of cells are expressing wildtype ubiquitin, whereas the bottom panel of cells are expressing Ser57Asp phosphomimetic ubiquitin. As in (**A**), cells are stably expressing Vph1-mCherry (red), a marker of the vacuolar membrane. (**C and D**) Analysis of yeast growth in the presence of canavanine. (**E**) Slot blot analysis of FLAG-ubiquitin levels following a galactose induction/glucose repression. (**F–J**) Quantitation of slot blot experiments (as shown in E, n = 3) from yeast cells expressing catalytic active or catalytic dead Hse1-UL36 fusion (**F and G**) or catalytic active Hse1-UL36 in the presence of wildtype or Ser57Asp phosphomimetic ubiquitin (**H and I**). Data were averaged over multiple experiments (**G and I**) (n = 3, error bars indicate standard deviation) and trend line regression was used to calculate ubiquitin half life (**J**).

DOI: https://doi.org/10.7554/eLife.29176.019

The following figure supplements are available for figure 5:

**Figure supplement 1.** Ser57Asp phosphomimetic ubiquitin does not alter the rate of Rsp5-mediated conjugation *in vitro*.

DOI: https://doi.org/10.7554/eLife.29176.020

**Figure supplement 2.** Ser57Asp phosphomimetic mutation does not alter binding of ubiquitin to ESCRT-I or ESCRT-II *in vitro*.

DOI: https://doi.org/10.7554/eLife.29176.021

*Figure 5 continued on next page*

*Figure 5 continued*

**Figure supplement 3.** DUB bypass by Ser57Asp phosphomimetic ubiquitin requires an in-tact ESCRT pathway while the autophagy pathway is dispensable.
DOI: https://doi.org/10.7554/eLife.29176.022
**Figure supplement 4.** Bypass of a Can1-DUB fusion by Ser57Asp phosphomimetic ubiquitin.
DOI: https://doi.org/10.7554/eLife.29176.023
**Figure supplement 5.** Phenotypic analysis of yeast cells expressing Leu73Pro ubiquitin.
DOI: https://doi.org/10.7554/eLife.29176.024

mutant cells exhibit increased rate of ubiquitin degradation (*Figure 6—figure supplement 1A and B*). Importantly, in Δ*doa4* mutant cells Ser57Asp and Ser57Ala mutations do not modify the rate of ubiquitin turnover (*Figure 6—figure supplement 1C and D*) – suggesting that the regulation of ubiquitin metabolism by Ser57 phosphorylation is Doa4-dependent. Taken together, our results confirm the role of Doa4 as a major determinant of ubiquitin turnover and indicate that ubiquitin phosphorylation at Ser57 can facilitate bypass of Doa4, and possibly other endosomal DUB activities, to promote ubiquitin degradation by delivery to the vacuole lumen.

## Most ubiquitin degradation in yeast is mediated by MVB sorting to the vacuole

Although our findings indicate that Ser57 phosphorylation plays an important role in the regulation of ubiquitin homeostasis, surprisingly little is known about mechanisms of ubiquitin degradation (*Kimura and Tanaka, 2010*; *Shabek and Ciechanover, 2010*; *Shabek et al., 2007*). In order to better understand mechanisms of ubiquitin degradation, we decided to assess the relative contribution of major protein degradation pathways to ubiquitin turnover. To our surprise, inhibition of the proteasome had minimal affect on ubiquitin half-life in the cell, at least in the culture conditions tested (*Figure 7A*). To test the role of vacuolar degradation in ubiquitin turnover, we measured ubiquitin half-life in a series of yeast mutants deleted for vacuolar proteases and found that strains lacking either Pep4 or Prb1 exhibited significant stabilization of ubiquitin (*Figure 7B*, *Figure 7—figure supplements 1* and *2*). These results indicate that most ubiquitin in the cell is degraded in the vacuole. Importantly, we found that deletion of Vps23 (ESCRT-I) or Snf7 (ESCRT-III) also significantly abrogated ubiquitin turnover (*Figure 7C and D*, *Figure 7—figure supplements 1* and *3*), indicating that most ubiquitin is delivered to the vacuole lumen for degradation by the MVB pathway. To visualize how abrogation of the ESCRT pathway impacts the subcellular distribution of ubiquitin, we expressed GFP-ubiquitin in wild-type and Δ*snf7* cells, chased each population following label with FM4-64 dye to label the limiting membrane of the vacuole, and mixed the two populations just prior to imaging in order to generate a side-by-side comparison of ubiquitin localization in the two backgrounds. In wild-type cells, we observed GFP-ubiquitin localized to many subcellular structures including the nucleus, vacuole, cytosol, plasma membrane, and various cytosolic punctae (*Figure 7E*, WT-labeled cells). In contrast, we found that ubiquitin accumulated primarily in the E-compartment (an aberrant endosomal structure) of Δ*snf7* cells (*Figure 7E*), suggesting that a significant amount of cellular ubiquitin transits through the MVB pathway. We conclude that – at least under the culture conditions tested in these experiments – there is substantial ubiquitin flux through the endocytic pathway, making it an ideal point of regulation for ubiquitin degradation.

## Discussion

For over a decade, post-translational modifications of ubiquitin have been detected in phosphoproteomic analysis of cells from multiple eukaryotic species (*Olsen et al., 2006*; *Peng et al., 2003*; *Rikova et al., 2007*; *Swaney et al., 2013*; *Villén et al., 2007*) but the functional significance of these modifications has only recently come into focus. Our data reveal that phosphorylation of ubiquitin at the Ser57 position is regulated by Ppz phosphatases in yeast, and elevated Ser57 phosphorylation in *ppz* mutants contributes to ubiquitin deficiency as well as other phenotypes. Furthermore, the data presented suggests that Ser57 phosphorylation of ubiquitin promotes both endocytic trafficking and ubiquitin degradation – two *in vivo* effects that can be explained by decreased susceptibility to cleavage by endosomal deubiquitylases as observed *in vitro* for Doa4. Finally, we show that vacuolar

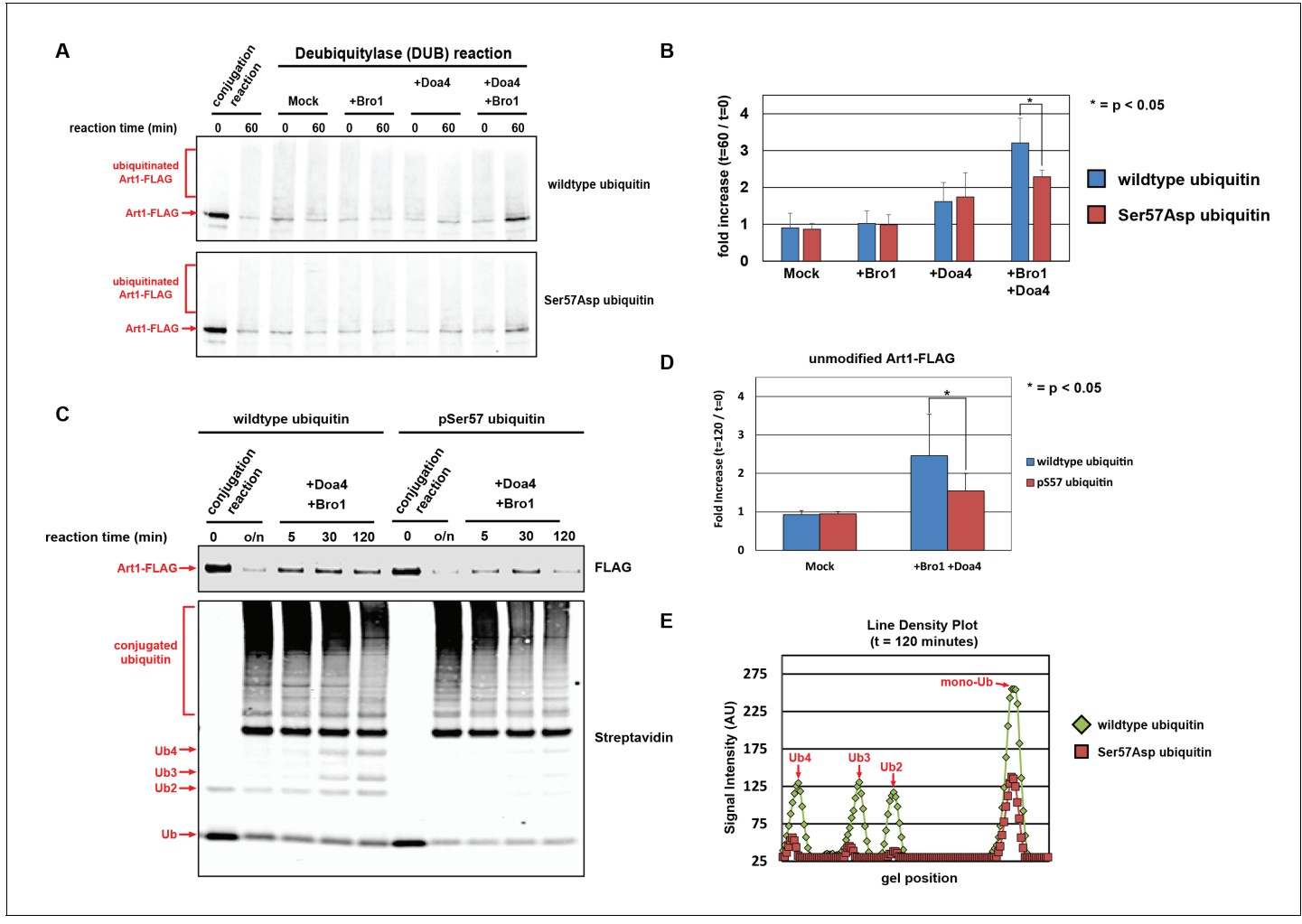

**Figure 6.** Ubiquitin phosphorylated at Ser57 is less susceptible to cleavage by Doa4. (**A**) Analysis of *in vitro* reconstituted Doa4 activity toward a substrate (Art1-3xFLAG, 50 nM) that has been conjugated to either wild-type or Ser57Asp phosphomimetic ubiquitin (see *Figure 5—figure supplement 1*). Deubiquitylation reactions were performed for 60 min in the presence of 40 nM 6xHIS-Doa4 (full length) and 3.2 μM Bro1 (C-terminal fragment with amino acids 388–844), where indicated. (**B**) Analysis of triplicate experiments as performed in (**A**). Doa4 deubiquitylase activity was measured as a function of the increase in intensity of the unmodified Art1 band relative to the start of the reaction. * indicates a statistically significant difference (p<0.05). (**C**) Analysis of *in vitro* reconstituted Doa4 activity toward a substrate (Art1-3xFLAG, 50 nM) that has been conjugated to either wild-type or Ser57 phosphorylated synthetic ubiquitin (Ubiquigent, Dundee, Scotland). Deubiquitylation reactions were performed in the presence of 40 nM 6xHIS-Doa4 (full length) and 3.2 μM Bro1 (C-terminal fragment with amino acids 388–844), where indicated. (**D**) Analysis of triplicate experiments as performed in (**C**). Doa4 deubiquitylase activity was measured as a function of the increase in intensity of the unmodified Art1 band relative to the start of the reaction. (**E**) Line density plots for the t = 120 min time point (**C**) were generated using ImageJ. * indicates a statistically significant difference (p<0.05).

DOI: https://doi.org/10.7554/eLife.29176.025

The following figure supplement is available for figure 6:

**Figure supplement 1.** Doa4 is a key determinant of cellular ubiquitin metabolism.
DOI: https://doi.org/10.7554/eLife.29176.026

degradation is the primary pathway for ubiquitin turnover in yeast cells, underscoring the Doa4-mediated recycling of ubiquitin as a critical point of regulation for global ubiquitin metabolism. Thus, we propose that Ser57 phosphorylation of ubiquitin can function as a regulatory switch to control ubiquitin recycling along the endocytic route, although the low observed stoichiometry of this modification suggests that this mode of regulation may occur transiently and only affect limited pools of ubiquitin (e.g. on the endosome).

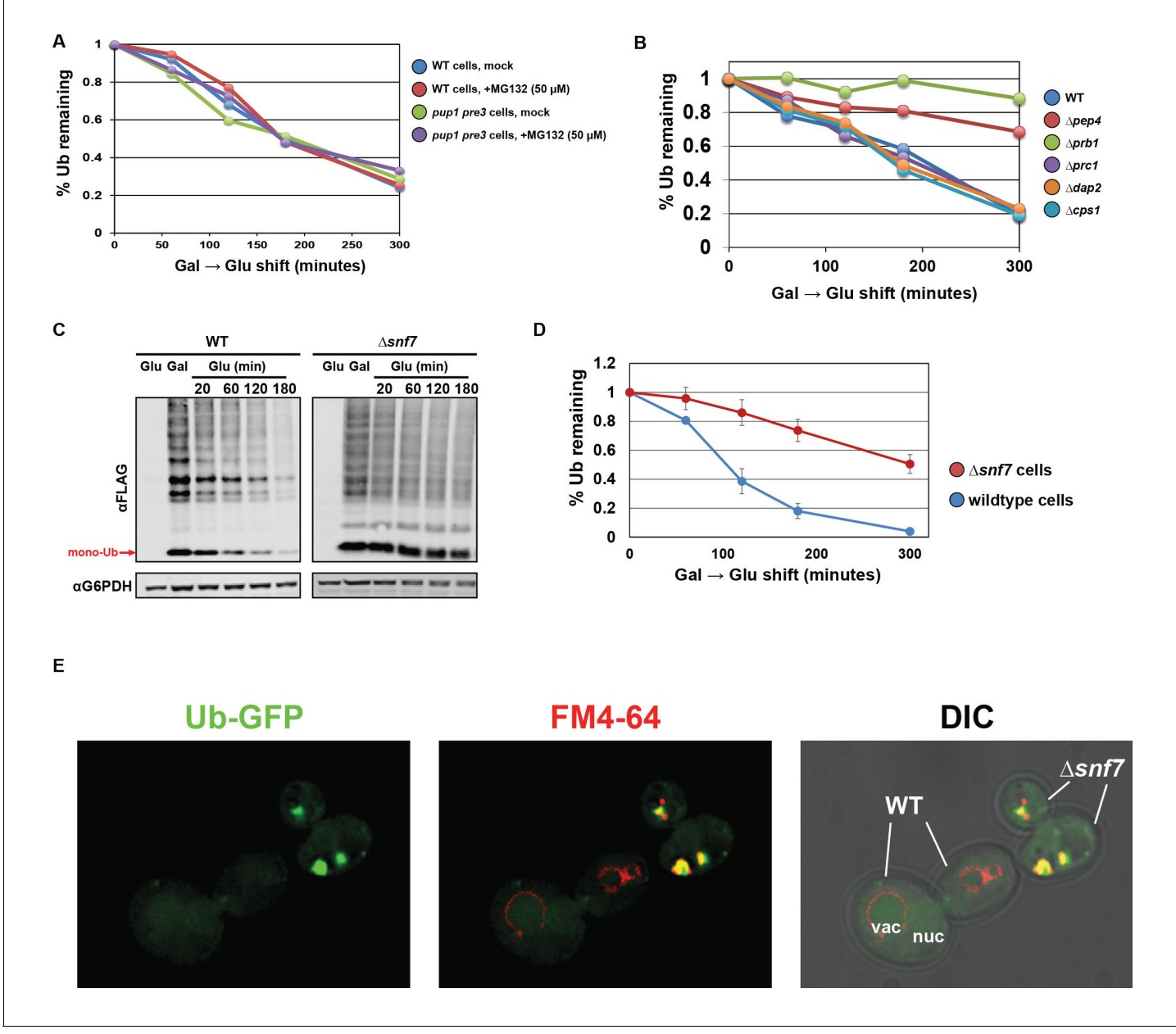

**Figure 7.** Ubiquitin is degraded in the vacuole via the ESCRT pathway. (**A**) Analysis of ubiquitin half-life following galactose induction/glucose repression in wild-type or *pup1pre3* mutant yeast cells (strain background WCG4A) that have been mock-treated or treated with 50 µM MG-132, a proteasome inhibitor. (**B**) Analysis of ubiquitin half-life following galactose induction/glucose repression in wild-type or vacuole protease mutant yeast cells (strain background BY4742). (**C**) SDS-PAGE and immunoblot analysis of lysates collected from yeast cells following a galactose induction/glucose repression experiment to measure ubiquitin half-life. (**D**) Quantitation of slot blot analysis of FLAG-ubiquitin levels following a galactose induction/ glucose repression (n = 4). Error bars indicate standard deviation. (**E**) Fluorescence microscopy imaging of GFP-tagged ubiquitin (green) in cells where FM4-64 (red) has been chased to label the vacuole membrane. Wild-type and Δ*snf7* cells were mixed prior to imaging, with the field showing both wild-type cells and Δ*snf7* cells, which are distinguished by the presence of an E-compartment.

DOI: https://doi.org/10.7554/eLife.29176.027

The following figure supplements are available for figure 7:

**Figure supplement 1.** Slot blot analysis of FLAG-ubiquitin levels following a galactose induction/glucose repression.
DOI: https://doi.org/10.7554/eLife.29176.028
**Figure supplement 2.** Quantitation of slot blot analysis of FLAG-ubiquitin levels following a galactose induction/glucose repression.
DOI: https://doi.org/10.7554/eLife.29176.029
**Figure supplement 3.** Quantitation of slot blot analysis of FLAG-ubiquitin levels following a galactose induction/glucose repression.

*Figure 7 continued on next page*

*Figure 7 continued*

DOI: https://doi.org/10.7554/eLife.29176.030

## Phosphorylation as an emerging mechanism of ubiquitin regulation

Recent biochemical and structural studies have provided key insights that suggest how Ser65 phosphorylation of ubiquitin could alter its biochemical properties in conjugation, recognition, and deubiquitylation (*Ordureau et al., 2014*; *Swaney et al., 2015*; *Wauer et al., 2015b*). In response to mitochondrial stress, Pink1 kinase accumulates on mitochondrial membranes and phosphorylates ubiquitin at Ser65. Ser65 phospho-ubiquitin binds with high affinity to Parkin, increasing its retention time and processivity on mitochondrial membranes (*Ordureau et al., 2015*; *Ordureau et al., 2014*; *Wauer et al., 2015b*) and inducing a conformational change in the RING1 domain that displaces Parkin's N-terminal Ubl domain, thus relieving autoinhibition (*Wauer et al., 2015a*). *In vitro*, Ser65 phosphorylation of ubiquitin resulted in defects in E2 charging, certain E3 conjugation reactions, and broadly inhibited processing by deubiquitylating enzymes (*Huguenin-Dezot et al., 2016*; *Swaney et al., 2015*; *Wauer et al., 2015b*). Indeed, given that phosphorylation at Ser65 can result in β-strand slippage causing the C-terminus of the protein to retract, it is not surprising Ser65 phosphorylation of ubiquitin significantly impacts its behavior in conjugation and de-conjugation reactions (*Swaney et al., 2015*; *Wauer et al., 2015b*). In contrast to Ser65, Ser57 is located on a surface loop and in the absence of structural data it is unclear how phosphorylation at this position would impact the overall structure of ubiquitin. Our NMR analysis of the Ser57Asp phosphomimetic suggests that structural perturbations are localized to the site of mutation (*Figure 3E and F*), although we cannot exclude the possibility that more substantial structural perturbations occur when Ser57 is phosphorylated. Furthermore, although Ser65 phosphomimetics were reported to activate parkin in cells (*Kane et al., 2014*; *Koyano et al., 2014*) they are unable to stimulate parkin activity *in vitro* (*Ordureau et al., 2015*) – underscoring the possibility at the Ser57Asp phosphomimetic ubiquitin does not necessarily phenocopy the effects of increased Ser57 phosphorylated ubiquitin. Thus, additional structural and biochemical characterization of Ser57-phosphorylated ubiquitin will be required in order to better understand how phosphorylation at the Ser57 position impacts the structure and function of ubiquitin.

Our *in vitro* analysis of Ser57Asp binding to ESCRT-I and ESCRT-II indicates UBD capture in those complexes is unaffected by phosphomimetic mutation at that position. This is not surprising, since most known ubiquitin interactions with UBDs occur via a hydrophobic patch (Leu8, Ile44, Val70) located on the opposite surface of ubiquitin (*Dikic et al., 2009*; *Komander and Rape, 2012*). Indeed, the Ser57 position does not appear to correspond to any surface features of ubiquitin that are known to participate in interactions with UBDs or DUBs, and thus it is unclear how depositing a phosphate group at this site would affect recognition by deubiquitylases. In the context of the ESCRT pathway, multiple UBDs have been described in ESCRT-0, ESCRT-I, ESCRT-II as well as Bro1 (*Shields et al., 2009*; *Shields and Piper, 2011*). Although the precise function of each UBD in this pathway is unclear, it is proposed that these UBDs function primarily in cargo capture and sorting for delivery to nascent ILVs as they form (*Henne et al., 2011*; *Ren and Hurley, 2010*; *Wollert and Hurley, 2010*). However, it is possible that different UBDs along the ESCRT pathway may have different functions – and it is tempting to speculate that one or more UBDs may be involved in positioning ubiquitin for recognition by Doa4 in order to be recycled prior to ILV scission. Because structural information on Doa4 is lacking and precise determinants of ubiquitin recognition by Doa4 in the context of ESCRT sorting are still unclear, further investigation will be required to determine how Ser57 phosphorylation of ubiquitin inhibits deconjugation by Doa4.

Although our study establishes a relationship between Ppz phosphatases and ubiquitin phosphorylation at the Ser57 position, additional biochemical studies will be required to determine if Ser57 phosphorylated ubiquitin is a direct substrate of Ppz phosphatases. This may be challenging since Ppz phosphatases are poorly characterized and appear to function in multi-subunit complexes but direct substrates have not been previously reported (*Posas et al., 1993*; *Ruiz et al., 2004*; *Sakumoto et al., 2002*). It is possible that ubiquitin is not a direct substrate of Ppz phosphatases, and the elevated levels of pSer57 ubiquitin in *ppz* mutants may be a cellular response to altered ubiquitin homeostasis. Indeed, other potential Ppz substrates identified in this study (*Figure 1—*

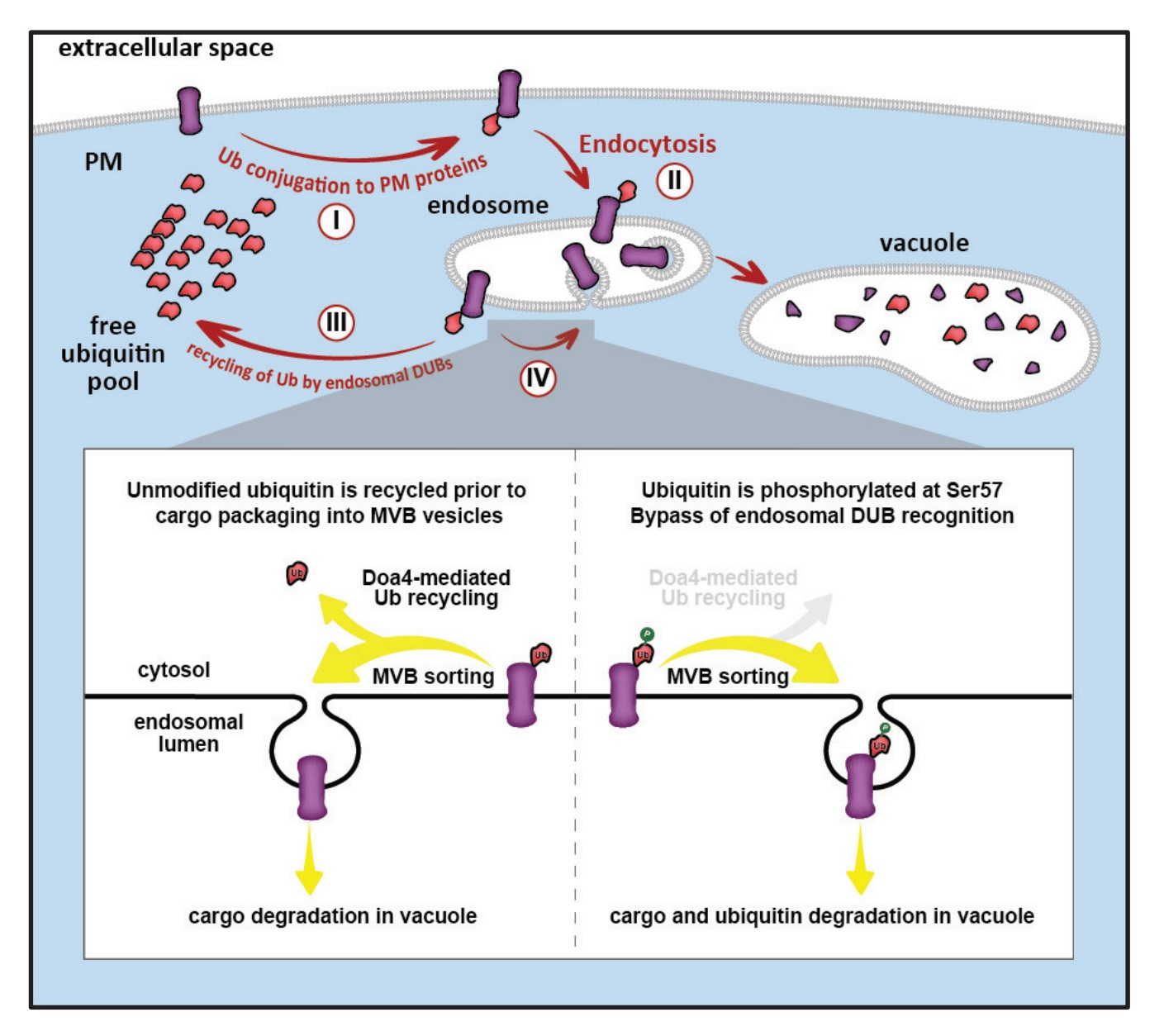

**Figure 8.** Model for ubiquitin flux through the endocytic pathway, illustrating how phosphorylation of ubiquitin at Ser57 can alter ubiquitin metabolism by limiting recycling of ubiquitin and promoting its delivery to the vacuole for degradation. (I) Plasma membrane proteins can be conjugated to ubiquitin, resulting in sorting and internalization by endocytosis (II). Ubiquitinated cargos on the limiting membrane of the endosome can be deubiquitylated (III) resulting in recycling of both cargo and ubiquitin, or they can be captured and sorted by ESCRT machinery into vesicles that bud into the lumen of the endosome. (IV) In the final stages of the ESCRT pathway, ubiquitin can be recycled prior packaging of cargo into vesicles by Doa4, but this mode of ubiquitin recycling is abrogated by phosphorylation at the Ser57 position of ubiquitin (inset).
DOI: https://doi.org/10.7554/eLife.29176.031

figure supplement 1 and Figure 1—source data 1) include Ypl191c, an uncharacterized member of the MINDY family of deubiquitylases (Abdul Rehman et al., 2016), and ECM30, which associates with the Ubp15 deubiquitylase. The potential regulation of these DUBs by Ppz phosphatases may explain why Ser57 phosphorylation of ubiquitin only partially accounts for the ubiquitin deficiency in *ppz* mutants (Figure 2A and B). Further experimentation will be required to understand additional layers of regulation that may contribute to the ubiquitin homeostasis phenotypes observed in *ppz*

mutant cells. Additionally, it will be important to identify yeast kinases that phosphorylate ubiquitin at the Ser57 position. Based on the data reported here, we hypothesize that Ser57 ubiquitin kinases in yeast might be involved in the management of ubiquitin homeostasis.

## Ubiquitin modification as a novel mechanism for regulation of membrane traffic

The current study demonstrates that Ser57Asp ubiquitin is associated with increased rates of endocytic trafficking (*Figure 4*) which can be explained, at least in part, by a DUB bypass mechanism (*Figure 5* and *Figure 6*). It is well-established that endosomal DUBs can remove ubiquitin from endocytic substrates to promote their recycling back to the plasma membrane (*Alwan and van Leeuwen, 2007*; *Clague et al., 2012*; *Clague and Urbé, 2006*). Additionally, cargo that have been sorted into the MVB pathway can encounter DUBs recruited by ESCRT-III which promote recycling of ubiquitin just prior to cargo delivery into intraluminal vesicles (*Amerik et al., 2000*; *Dupré and Haguenauer-Tsapis, 2001*; *Henne et al., 2011*). In yeast, Doa4 appears to be the primary DUB responsible for recycling ubiquitin during MVB biogenesis, consistent with the fact that Δ*doa4* mutant cells exhibit ubiquitin deficiency (*Swaminathan et al., 1999*) and accelerated turnover (*Figure 6— figure supplement 1*). Doa4 recruitment to endosomes is mediated by an interaction with Bro1, a highly conserved yeast homolog of the human ALIX protein which is essential for MVB sorting and also plays a role in Doa4 activation on endosomes (*Luhtala and Odorizzi, 2004a*; *Richter et al., 2007*). Rfu1, a protein that has been proposed to function as an inhibitor of Doa4 activity (*Kimura et al., 2009*), is also recruited to endosomes by interactions with Bro1 (*Kimura et al., 2014*). Our data indicate that the Ser57 position of ubiquitin plays an important role in recognition by Doa4. Based on these results, we propose that Doa4 activity is an important point-of-regulation to control both rates of endocytic flux and ubiquitin turnover, especially in conditions of stress. In addition to the established regulatory roles for Bro1 (an activator) and Rfu1 (an inhibitor), we propose that Doa4 activity can also be regulated by ubiquitin phosphorylation at the Ser57 position. Additional studies will be required to determine if Ser57 phosphorylation of ubiquitin prevents Doa4 recognition directly or indirectly via regulation of Rfu1 or Bro1.

We also considered the possibility that Ser57 phosphorylation of ubiquitin could accelerate endocytic trafficking by increasing the rate of Rsp5-mediated ubiquitin conjugation or by altering affinity for UBDs in the ESCRT pathway; however, we found no evidence that Ser57Asp ubiquitin exhibited either of these properties (*Figure 5—figure supplement 1* and *Figure 5—figure supplement 2*). Despite these findings, we cannot exclude the possibility that Ser57 phospho-ubiquitin might exhibit increased rate of Rsp5-mediated conjugation or increased altered affinity for UBDs in the ESCRT pathway, which could contribute to the accelerated endocytic trafficking phenotype. The possibility that post-translational modifications of ubiquitin regulate the efficiency of endocytosis, cargo sorting on endosomes, or other membrane protein sorting and trafficking events in the cell represents a novel mode of regulation that will require further exploration in future studies.

## Harvesting the reaper: vacuolar trafficking and ubiquitin metabolism

Ubiquitin homeostasis is poorly understood but clearly involves complex sensing and regulation for the control of both ubiquitin expression and degradation. Most mutations known to alter ubiquitin homeostasis occur in either deubiquitylating enzymes or genes encoding the ubiquitin conjugation machinery. To our knowledge, the current study provides the first example of a phosphatase – or any signaling protein – that is required for proper ubiquitin homeostasis. We predict that signaling components upstream of Ppz phosphatases may sense ubiquitin levels and, by regulating the activity of Ppz phosphatases and possibly other factors involved in ubiquitin recycling, control ubiquitin metabolism. It is noteworthy that we observe the greatest amount of ubiquitin Ser57 phosphorylation in *ppz* mutant cells grown in minimal media (*Figure 1A*). Interestingly, we normally do not detect high levels of Ppz phosphatase expression in yeast cells grown to mid-log phase, whereas higher levels of expression can be detected in stationary phase yeast cultures (data not shown). These observations indicate that Ppz phosphatases may function as part of a signaling mechanism that operates to manage ubiquitin levels in response to changes in nutrient availability.

Although little is known about mechanisms of ubiquitin degradation, previous studies have reported that ubiquitin can be degraded by the proteasome (*Shabek and Ciechanover, 2010*;

*Shabek et al., 2009*; *Shabek et al., 2007*) and, in the case of Δ*doa4* mutant cells, by the vacuole (*Amerik et al., 2000*; *Dupré and Haguenauer-Tsapis, 2001*; *Swaminathan et al., 1999*). We were surprised to find that proteasome activity appeared to be largely dispensable for ubiquitin degradation while defects in vacuolar degradation or MVB sorting led to significantly reduced rates of ubiquitin turnover (*Figure 7*). These findings suggest that ESCRT-mediated vacuolar trafficking is the major mechanism of ubiquitin degradation in yeast, at least under the conditions tested in this study. Indeed, the observation that ubiquitin accumulates significantly in E-compartments of *escrt* mutants (*Figure 7E*) is an indication that ubiquitin flux through the endocytic pathway is significant, making endosomes an ideal location for regulating whether ubiquitin is recycled or degraded (*Figure 8*). Ultimately, understanding how cells sense and manage ubiquitin homeostasis may lead to therapeutic targets and strategies for restoration of ubiquitin pools during aging or in disease contexts such as neurodegeneration where ubiquitin is depleted.

# Materials and methods

**Key resources table**

| Reagent type (species) or resource | Designation | Source or reference | Identifiers | Additional information |
|---|---|---|---|---|
| Strain background (*S. cerevisiae*) | SEY6210.1 | S. Emr Lab; *Robinson et al. (1988)* | | WT: Mat a leu2-3,112 ura4-52 his3-Δ200 trp1-Δ901 lys2-801 suc2-Δ9 |
| Strain background (*S. cerevisiae*) | SUB280 | D. Finley Lab; *Finley et al. (1994)* | | MATa, lys2-801, leu2-3, 112, ura3-52, his3-Δ200, trp 1–1, ubi1-Δ1::TRP1, ubi2-Δ2::ura3, ubi3-Δub-2, ubi4-Δ2::LEU2 [pUB39 Ub, LYS2] [pUB100, HIS3] |
| Strain background (*S. cerevisiae*) | BY4742 | Dharmacon (Lafayette, CO) | | MATα his3Δ1 leu2Δ0 lys2Δ0 ura3Δ0 |
| Strain background (*S. cerevisiae*) | WCG4a | B. Tansey Lab; *Howard et al. (2012)* | | MATa ura3 leu2-3,112 his3-11,15 |
| Strain background (*S. cerevisiae*) | WT | this paper | | SEY6210.1; MATa, TRP::HTF-ubi2, TRP::HTF-ubi3, arg4Δ::Kan |
| Strain background (*S. cerevisiae*) | Δppz1,2 | this paper | | SEY6210.1; MATa, TRP::HTF-ubi2, TRP::HTF-ubi3, arg4Δ::Kan, ppz1Δ::HIS, ppz2Δ::TRP |
| Strain background (*S. cerevisiae*) | WT-Ub WT;S57A;S57D | this paper | | SUB280; RPS31 (UB-WT/S57A/S57D) |
| Strain background (*S. cerevisiae*) | Δppz1,2 | this paper | | SUB280; ppz1::KAN, ppz2::NAT |
| Strain background (*S. cerevisiae*) | Δppz1,2-Ub WT;S57A;S57D | this paper | | SUB280; ppz1::KAN, ppz2::NAT, RPS31 (UB-WT/S57A/S57D) |
| Strain background (*S. cerevisiae*) | WT | this paper | | SEY6210.1; Mup1-pHluorin::KanMX |
| Strain background (*S. cerevisiae*) | Δppz1,2 | this paper | | SEY6210.1; Mup1-pHluorin::KanMX, ppz1Δ::HIS, ppz2Δ::TRP |
| Strain background (*S. cerevisiae*) | WT-Ub WT;S57A;S57D | this paper | | SEY6210.1; RPS31-HTF (UB-WT/S57A/S57D) |
| Strain background (*S. cerevisiae*) | WT | this paper | | SEY6210.1; Mup1::HTP-TRP |
| Strain background (*S. cerevisiae*) | Δppz1,2 | this paper | | SEY6210.1; Mup1::HTP-TRP, ppz1Δ::HIS, ppz2Δ::TRP |
| Strain background (*S. cerevisiae*) | Vph1-mCherry | this paper | | SEY6210.1; Vph1-mCherry |
| Strain background (*S. cerevisiae*) | Hse1-DUB | this paper | | SEY6210.1; Hse1-DUB (HA-UL36)::NAT |
| Strain background (*S. cerevisiae*) | Δsnf7 | this paper | | SEY6210.1; Hse1-DUB (HA-UL36)::NAT, snf7::His |

*Continued on next page*

*Continued*

| Reagent type (species) or resource | Designation | Source or reference | Identifiers | Additional information |
|---|---|---|---|---|
| Strain background (*S. cerevisiae*) | Δvps23 | this paper | | SEY6210.1; Hse1-DUB (HA-UL36)::NAT, vps23::His |
| Strain background (*S. cerevisiae*) | Δatg5 | this paper | | SEY6210.1; Hse1-DUB (HA-UL36)::NAT, atg5::His |
| Strain background (*S. cerevisiae*) | Δatg8 | this paper | | SEY6210.1; Hse1-DUB (HA-UL36)::NAT, atg8::His |
| Strain background (*S. cerevisiae*) | Can1-DUB | this paper | | SEY6210.1; Can1-DUB (HA-UL36)::NAT |
| Antibody (Mouse-Monoclonal ANTI-FLAG(R) M2 antibody) | α-Flag | Sigma (St. Louis, MO) | AB_262044 | dilution 1:1000 |
| Antibody (Mouse-MAB1510) | α-Ubiquitin | Millipore (Burlington, MA) | AB_2180556 | dilution 1:1000 |
| Antibody (Rabbit-G6PDH) | α-G6PDH | Sigma (St. Louis, MO) | AB_258454 | dilution 1:10000 |
| Antibody (Rabbit-Ubiquitin-pS57) | α-pSer57 Ub | Gift from B. Brasher | | dilution 1:1000 |
| Antibody (IRDye 680RD-Goat anti-mouse) | | Licor (Lincoln, NE) | Cat.# 926–68070 | dilution 1:10000 |
| Antibody (IRDye 800CW-Goat anti-rabbit) | | Licor (Lincoln, NE) | Cat.# 926–32211 | dilution 1:10000 |
| Chemical compound, drug (Cycloheximide) | CHX | Sigma (St. Louis, MO) | Cat. # C7698 and C1988 | |
| Chemical compound, drug (L-Canavanine) | Canavanine | Sigma (St. Louis, MO) | Cat. # C1625 | |
| Chemical compound, drug (DL-2-Aminoadipic acid) | DL-2-Aminoadipic acid | Sigma (St. Louis, MO) | Cat. # A0637 | |
| Chemical compound, drug (FM4-64) | FM4-64 | Invitrogen (Carlsbad, CA) | Cat. # T3166 | |
| Chemical compound, drug (MG132) | MG132 | Apexbio (Houston, TX) | Cat. # A2585 | |
| Expression vectors for yeast (pRS416) | | *Sikorski and Hieter (1989)* | | |
| Expression vectors for yeast (pRS416, RPS31 (UB-WT/S57A/S57D)) | Ub-WT;S57A;S57D | this paper | | |
| Expression vectors for yeast (pRS416, pADH-RPS31 (UB-WT/S57A/S57D)) | (ADH1) Ub-WT; S57A;S57D | this paper | | |
| Expression vectors for yeast (pRS416, pTDH-RPS31 (UB-WT/S57A/S57D)) | (TDH3) Ub-WT; S57A;S57D | this paper | | |
| Expression vectors for yeast (pRS416, PPZ1-HTF) | PPZ1-Flag | this paper | | |
| Expression vectors for yeast (pRS416, PPZ1-R451L-HTF) | PPZ1-R451L-Flag | this paper | | |
| Expression vectors for yeast (pRS416, RPS31 (WT/S57A/S57D), pTEF1-HTF) | (TEF1) Ub-WT; S57A;S57D | this paper | | |
| Expression vectors for yeast (pRS416, pGAL10-HTF (UB-WT/S57A/S57D)) | (GAL10) Ub-WT; S57A;S57D | this paper | | |
| Expression vectors for yeast (pRS416, MUP1-GFP) | MUP1-GFP | this paper | | |

*Continued on next page*

*Continued*

| Reagent type (species) or resource | Designation | Source or reference | Identifiers | Additional information |
|---|---|---|---|---|
| Expression vectors for yeast (pRS416, ART1-HTF) | ART1-Flag | this paper | | |
| Expression vectors for yeast (pRS416, pADH-RPS31, L73P) | Ub-L73P | this paper | | |
| Expression vectors for yeast (pRS416, GFP-Ub) | GFP-Ub | this paper | | |
| Software | Adobe Illustrator CS5.1 | Version 15.1.0 | | |
| Software | ImageJ | NIH | SCR_003070 | |
| Software | Licor image studio | Licor (Lincoln, NE) | SCR_015795 | |

## Yeast strains and growth conditions

The SEY6210 strain background (*MATα leu2-3,112 ura3-52 his3-Δ200 trp1-Δ901 lys2-801 suc2-Δ9*) was used for most experiments. The SUB280 yeast strain background (*MATa lys2-801 leu2-3,112 ura3-52 his3-Δ200 trp 1–1 ubi1-Δ1::TRP1 ubi2-Δ2::ura3 ubi3-Δub-2 ubi4-Δ2::LEU2 [pUB39 Ub, LYS2] [pUB100, HIS3]*) (generously supplied by D. Finley and M. Boselli) (*Spence et al., 1995*), which lacks all chromosomal ubiquitin genes and expresses ubiquitin from a single counter-selectable plasmid, was used for experiments that required expression of ubiquitin from a single source. Using this background, we shuffled in pRS416 (*URA3*) vectors containing the *RPS31* ubiquitin gene (or indicated mutant variants with indicated promoters) and selected for loss of pUB39 (which expresses ubiquitin from the *CUP1* promoter) by counter-selection on plates containing α-aminoadipate (*Sikorski and Boeke, 1991*; *Sloper-Mould et al., 2001*). All yeast strains generated in this manner were confirmed by counter-selection with 5-FOA – which is toxic in the presence of the pRS416 (*URA3*) plasmid – to ensure that ubiquitin expression from the pRS416 plasmid was essential. Unless otherwise indicated, all standard yeast culture conditions and canavanine plating assays were performed as previously described (*Lin et al., 2008*).

## Antibodies and analysis of cellular protein expression levels

Unless otherwise indicated, yeast lysates were prepared by growing cultures according to the indicated conditions, pelleting 5 $OD_{600}$ equivalents of cells and precipitating by addition of 10% trichloroacetic acid (TCA). Precipitates were washed in acetone, aspirated, dried under vacuum in a speed-vac, re-suspended in lysis buffer (150 mM NaCl, 50 mM Tris pH7.5, 1 mM EDTA, 1% SDS), and mechanically disrupted with acid-washed glass beads. Protein sample buffer (150 mM Tris pH 6.8, 6 M Urea, 6% SDS, 10% beta-mercaptoethanol, 20% Glycerol) was added and extracts were analyzed by SDS-PAGE and immunoblotting. Quantitative fluorescence imaging of immunoblots was performed using an Odyssey infrared imaging system (LI-COR Biosciences, Lincoln, NE). Antibodies used in this study include: α-FLAG (M2, Sigma, St. Louis, MO); α-G6PDH (Sigma, St. Louis, MO), α-ubiquitin (MAB1510, Millipore, Burlington, MA), and α-pSer57 phospho-ubiquitin (generously supplied by B. Brasher, BostonBiochem, Cambridge, MA).

## Quantitative proteomic analysis

Quantitative mass spectrometry analysis by SILAC was performed on yeast strains auxotrophic for lysine and arginine. Cells were grown to mid-log phase in the presence of either heavy or light isotopes (lysine and arginine) and affinity purification was performed as previously described (*MacGurn et al., 2011*; *Manford et al., 2012*). For phosphoproteome experiments, phosphopeptides were purified using IMAC chromatography as previously described (*Albuquerque et al., 2008*; *MacGurn et al., 2011*). Purified peptides were dried, reconstituted in 0.1% trifluoroacetic acid, and analysed by LC-MS/MS using an Orbitrap XL mass spectrometer. Database search and SILAC quantitation were performed using Sorcerer software (Sage-N, Milpitas, CA).

For absolute quantification of peptide abundance by mass spectrometry using isotopically labelled peptide standards, FLAG-tagged ubiquitin was affinity purified from yeast cultures grown to mid-log phase in YPD (OD600 = 0.7; ~700 OD equivalents). Cell lysates were generated by bead

beating in lysis buffer (50 mM Tris-HCl, pH 7.5, 0.2% NP-40, 150 mM NaCl with protease and phosphatase inhibitors). Affinity purification was performed by binding to anti-FLAG M2 affinity gel (Sigma, St. Louis, MO) for 2 hr at 4°C, washing beads three times with lysis buffer, and elution by boiling beads for 5 min in elution solution (50 mM Tris, pH 8.0, 1% SDS) followed by collection of the sample through a tip-column, sample reduction (10 mM DTT, 5 min at 95°C), and alkylation (20 mM iodoacetamide, room temperature). Samples were digested with 1 μg of trypsin overnight at 37°C and purified over a C18 column (Waters). Isotopically labelled standard peptides (21st Century Biochemicals, Marlboro, MA) were spiked into samples at different quantities (approximating peptide abundance observed in samples based on generation of a standard curves), separated over a microbore C-18 column (25 cm x 100 μm) using a 90-min aqueous to organic gradient, and analyzed by multiple reaction monitoring on a TSQ-Vantage mass spectrometer (Thermo Scientific, Rockford, IL). Data analysis and quantification of transitions were performed using Skyline software (MacCoss Lab).

## Analysis of ubiquitin turnover

For cycloheximide (CHX) chase experiments, yeast cultures were grown to mid-log phase and 50 μg/mL cycloheximide was added to the media at t = 0 and yeast samples were TCA precipitated at the indicated time points. For experiments using galactose induction/glucose repression to measure ubiquitin turnover, cells were grown to mid-log phase in media containing glucose (SCD synthetic media or YPD rich media), washed twice in glucose-free media, resuspended in media containing 2% galactose (SCG or YPG) and incubated at 26°C with shaking overnight. Following galactose induction, ubiquitin expression was repressed by washing cultures twice and resuspending in media containing 2% glucose and yeast samples were TCA precipitated at the indicated time points. Time course experiments were performed in triplicate using either SDS-PAGE with immunoblotting or slot blots, and band intensities were analyzed and quantified using ImageJ software.

## NMR analysis of ubiquitin structure

$^{15}$N-enriched wildtype and S57D ubiquitin was produced in Rosetta cells cultured in a M9 Minimal Media containing 0.5 g/L $^{15}$N-NH$_4$Cl as the sole nitrogen source. Cells were induced at an OD600 of 0.3 with 0.5 mM IPTG and allowed to express for 3 hr at 37°C. Cells were pelleted by centrifugation at 5000 x g for 10 min, washed in lysis buffer (20 mM ammonium acetate pH 5.1), pelleted again, and flash frozen. Prior to purification, the cell pellet was thawed on ice, resuspended in 15 mL of lysis buffer [20 mM ammonium acetate, complete protease inhibitors (Roche, Basel, Switzerland), lysozyme and 1 mM PMSF], and sonicated (20 min total, 20 s on and 30 s off). Cell lysates were cleared by centrifugation (20,000 rpm for 20 min at 4°C) and filtered through a 0.45 μM filter. Contaminants were precipitated by titrating acetic acid until the pH dropped to 4.8, then the precipitated protein was pelleted by centrifugation (20,000 rpm for 20 min at 4°C). The cleared lysate was run over a C4 proto column (Higgins Analytical, Inc., Mountain View, CA) using a Waters Delta 600 HPLC system. Ubiquitin containing fractions were buffer exchanged into NMR buffer (20 mM sodium phosphate pH 6.5, 50 mM NaCl) and concentrated by centrifugation to 100 μM. $^{15}$N-$^1$H HSQC experiments were performed at 25°C using a Bruker 600 MHz Avance AV-III spectrometer. Data were processed in NMRPipe and visualized using CARA.

## Fluorescence microscopy and trafficking assays

Yeast cells expressing fluorescent fusion proteins were grown to mid-log phase in synthetic media. Microscopy images were acquired using a DeltaVision Elite system (GE Healthcare) and processed using SoftWoRx software (GE Healthcare, Chicago, IL). FM4-64 was used as a marker to label the limiting membrane of the vacuole as previously described (*Vida and Emr, 1995*).

## Recombinant protein expression and purification

For bacterial expression, plasmids containing N-terminally tagged Ubc1, Bro1, and Art1 genes were transformed into Rosetta cells. 5 ml of LB/Ampicillin/chloramphenicol media were grown overnight at 37°C with a single colony from the transformation. The 5 ml starter cultures were added to 1 L of LB media with antibiotics and grown to OD$_{600}$ of approximately 0.6. At this point the cultures were induced with 0.5 mM IPTG and left to shake overnight at 18°C. Cells were harvested by

centrifugation at 6000 x g for 20 min. The pellets were washed with 50 ml of lysis buffer (50 mM NaPO$_4$, 300 mM NaCl, pH 8.0) per 1 L of culture then centrifuged at 2000 rpm for 30 min. Pellets were resuspended in 20 ml of lysis buffer with protease tablets and 1 mM PMSF. Cells were lysed by sonication (5 × 1 min, 50% duty) then spun down at 12,000 rpm for 20 min. For purification, lysates were applied to either Ni-NTA or Cobalt resin (Thermo Scientific, Rockford, IL) that had been equilibrated with lysis buffer containing 20 mM imidazole. The protein was eluted with lysis buffer containing 300 mM imidazole. Protein was concentrated and buffer exchanged to remove the imidazole. Cells containing an overexpression plasmid for Rsp5 with a Strep-tag were grown and lysed as described above except the lysis buffer used was 100 mM Tris pH 8, 150 mM NaCl, 1 mM EDTA. To purify, 10 ml of lysate was applied to a 0.2 ml Strep-Tactin column (IBA Lifesciences, Göttingen, Germany). The protein was eluted with 0.8 mM biotin.

### *In vitro* ubiquitin conjugation and deubiquitylation assays

For the conjugation reactions, 56 nM GST-Ube1 (Boston Biochem, Cambridge, MA, E-300), 0.77 mM His$_6$-Ubc1, 2.3 mM Ub or Ub S57D, 50 nM Art1-flag, and 60 nM Strep-RSP5 were incubated in conjugation buffer (40 mM Tris pH 7.5, 10 mM MgCl$_2$ and 0.6 mM DTT) with 1 mM ATP, at 30°C. Reactions were initiated by the addition of Strep-RSP5. Samples were removed at indicated time-points, boiled in 2x Laemmli sample buffer for 10 min, and analyzed by blotting for Art1-flag with flag antibody (Sigma, St. Louis, MO) using goat anti-mouse secondary (IRDye 680RD, LI-COR Biosciences, Lincoln, NE). Quantification was performed using ImageStudioLite software (LI-COR).

For deconjugation reactions, polyubiquitinated substrate (Art1-flag) was generated using the conjugation assay described above within one exception in that 0.1 mM ATP was used. The conjugation reaction was stopped by incubation with 0.75 units Apyrase in a 90 μl reaction at 30°C for 1 hr. For the deconjugation reaction, 40 nM His$_6$-Doa4 and 3.2 mM His$_6$-Bro1$^{388-844}$ was added to conjugated Art1-flag as well as 10X DUB buffer (0.5 M Hepes pH 7.4, 200 mM DTT, 3 mg/ml BSA) to 1X. Reactions were initiated by the addition of His$_6$-Doa4 and His$_6$-Bro1$^{388-844}$ protein. The reaction was incubated at room temperature. Samples were removed at indicated time-points and boiled in 2x Laemmli sample buffer. Art1-flag levels were determined as described in the conjugation assay.

## Acknowledgements

We are very grateful to D Finley, M Boselli, R Piper, W Tansey, C Howard, S Wente, T Dawson, T Graham, and P Xu for yeast strains and reagents. We are very grateful to B Brasher for supplying the α-pSer57 antibody used in these studies. We thank Dr. Markus Voehler for technical support for the NMR experiments. We acknowledge M Smolka and K Rose for helpful advice regarding technical aspects and analysis of quantitative proteomic data. We thank T Graham and W Tansey for critical reading of this manuscript. We are especially grateful to S Emr for critical feedback and sage advice. We are grateful to E MacGurn for assistance with graphic design. JMT is funded by NIH training grant T32 CA119925. SL was supported by NIH training grant T32 HL069765. ACE was supported by NIH training grant T32 CA009582. This research was supported by NIH grant R00 GM101077 (to JAM), NIH grant R01 GM118491 (to JAM) and by NIH grant R35 GM118089 (to WJC). The NMR instrumentation was supported in part by grants from the NSF (0922862), NIH (S10 RR025677) and Vanderbilt University matching funds. This research was conducted while JAM was an AFAR Research Grant recipient from the American Federation for Aging Research.

## Additional information

### Funding

| Funder | Grant reference number | Author |
| --- | --- | --- |
| National Institute of General Medical Sciences | R00 GM101077 | Jason A MacGurn |
| National Institute of General Medical Sciences | R01 GM118491 | Jason A MacGurn |
| American Federation for Aging Research | Research Grants for Junior Faculty | Jason A MacGurn |

| National Institute of General Medical Sciences | R35 GM118089 | Walter J Chazin |
| National Heart, Lung, and Blood Institute | T32 HL069765 | Sora Lee |
| National Cancer Institute | T32 CA119925 | Jessica M Tumolo |
| National Cancer Institute | T32 CA009582 | Aaron C Ehlinger |

The funders had no role in study design, data collection and interpretation, or the decision to submit the work for publication.

### Author contributions

Sora Lee, Conceptualization, Validation, Investigation, Methodology, Writing—review and editing; Jessica M Tumolo, Walter J Chazin, Conceptualization, Supervision, Investigation, Methodology, Writing—review and editing; Aaron C Ehlinger, Jason A MacGurn, Conceptualization, Supervision, Funding acquisition, Investigation, Methodology, Writing—original draft, Project administration, Writing—review and editing; Kristin K Jernigan, Conceptualization, Investigation, Methodology; Susan J Qualls-Histed, Pi-Chiang Hsu, Conceptualization, Resources, Investigation; W Hayes McDonald, Investigation, Methodology

### Author ORCIDs

Sora Lee, http://orcid.org/0000-0001-9514-8152
Jason A MacGurn, http://orcid.org/0000-0001-5063-259X

### Decision letter and Author response

Decision letter https://doi.org/10.7554/eLife.29176.033
Author response https://doi.org/10.7554/eLife.29176.034

## Additional files

### Supplementary files

• Transparent reporting form
DOI: https://doi.org/10.7554/eLife.29176.032

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
