## [Decision Letter]

Thank you for submitting your article "Ubiquitin turnover and endocytic trafficking in yeast are regulated by Ser57 phosphorylation of ubiquitin" for consideration by *eLife*. Your article has been favorably evaluated by Vivek Malhotra (Senior Editor) and three reviewers, one of whom, Wade Harper, is a member of our Board of Reviewing Editors. The following individual involved in review of your submission has agreed to reveal his identity: Richard Gardner (Reviewer #3).

The reviewers have discussed the reviews with one another and the Reviewing Editor has drafted this decision to help you prepare a revised submission.

Summary:

This work examines ubiquitin (UB) phosphorylation in budding yeast and in particular, the potential roles of pS57-UB. The authors find that the abundance of pS57 UB is increased when redundant phosphatase regulatory subunits ppz1/2 are deleted. Using strains expressing either phosphomimetics or non-phosphorylatable mutants in UB (S57A), the authors report that turnover of UB is accelerated when it is in the phospho-mimic form while the S57A mutant does not have a phenotype. In addition, the authors find that expression of the S57D phosphomimic alters the rate of endocytosis in a manner that depends on the deubiquitylating enzyme Doa4.

Essential revisions:

All of the reviewers find the paper to be potentially interesting and important. The role of UB phosphorylation in specific functions of UB biology is in its infancy. However, the reviewers expressed some concerns with particular aspects of the paper that need to be satisfactorily addressed. Although there was some differences in opinion about the relative importance of particular issues, all reviewers agree that the following issues should be addressed.

1) A major conclusion of the paper is that pS57-UB promotes UB turnover as well as endocytosis. This is based on the use of a phosphomimetic or the S57A mutant that is either the only UB form expressed or the one that is followed. The S57D mutant has a phenotype while the S57A doesn't. The absence of a phenotype with the S57A mutant is problematic since this would seem to suggest that phosphorylation at this site isn't absolutely required for the functions that are examined. This would suggest that the role of S57 phosphorylation in the phenotypes studied is minor.

The reviewers suggest three approaches to try to address this: a) The stoichiometry of S57 phosphorylation needs to be measured with and without ppz. This can be done using various approaches, but perhaps quantitative proteomics is the best approach. If the stoichiometry is low, you would need to accurately describe the implications of this on your conclusions. It may be the case that normally, a very small fraction of UB is phosphorylated on this site and that pool is more rapidly degraded during the execution of endocytosis.

b) Further analysis of the ppz1/2 deletion strain which has elevated S57 phosphorylated ubiquitin would strengthen the paper, as this seems a more natural and relevant condition than overexpression of the S57D mutant ubiquitin. For example, what does Mup1 trafficking look like in the ppz1/2 deletion strain (Figure 4), or does the ppz1/2 strain display the same bypass of Hse1-UL36 for Mup1-GFP trafficking (Figure 5) as the S57D mutant ubiquitin strain?

c) Depending on the results of points 1 and 2, you would need to be able to provide a more nuanced description of the results, as there is concern that some of the conclusions are too strong. In particular, the fact that much of the UB is degraded in the context of the S57D mutant is likely not to be physiological, since the stoichiometry is likely low compared to the apparent case of the phosphomimetic. So a much more nuanced set of conclusions should be made, depending on the outcome.

d) There is also an issue with the possibility that the phosphomimetic doesn't actually phenocopy the phosphoform. In the case of pS65 UB, it is clear from the literature that the phosphomimetic is not really a phosphomimetic biochemically, as pS65 UB can dramatically activate parkin in vitro but S65D UB has essentially no activity in such assays. Some discussion of this issue or qualifications in the text is likely needed.

2) Overall, the treatment of the mass spectrometry experiments is insufficiently described. It is unclear for example how error bars are being generated from a SILAC experiment apparently performed only once unless the error bars are coming from different peptides, but this can't really be done for single peptides by SILAC. A detailed description of how the statistics was done is needed.

3) There was some concern and discussion concerning the fact that the strains have overexpressed UB in some cases, although the reviewers understand the difficulties in making mutations in the endogenous loci. It might be useful to have blots with all the UB chains collapsed down to mono UB with USP2 in order to know what the actual UB levels are.

4) Quite a number of different yeast strains with different ways of expressing UB are used. Perhaps this should be explained.

---

## [Author Response]

Essential revisions:All of the reviewers find the paper to be potentially interesting and important. The role of UB phosphorylation in specific functions of UB biology is in its infancy. However, the reviewers expressed some concerns with particular aspects of the paper that need to be satisfactorily addressed. Although there was some differences in opinion about the relative importance of particular issues, all reviewers agree that the following issues should be addressed.1) A major conclusion of the paper is that pS57-UB promotes UB turnover as well as endocytosis. This is based on the use of a phosphomimetic or the S57A mutant that is either the only UB form expressed or the one that is followed. The S57D mutant has a phenotype while the S57A doesn't. The absence of a phenotype with the S57A mutant is problematic since this would seem to suggest that phosphorylation at this site isn't absolutely required for the functions that are examined. This would suggest that the role of S57 phosphorylation in the phenotypes studied is minor.

We generally agree with this assessment but we would also point out that the Ser57Ala mutant does suppress or partially suppress some of the *ppz* mutant phenotypes (Figure 2) and partially suppresses the *ppz* ubiquitin deficiency (Figure 2). These results indicate some functional contribution of Ser57 phosphorylation to the *ppz* mutant phenotypes – however, the results also indicate that the ubiquitin deficiency phenotype is not entirely attributable to Ser57 phosphorylation, suggesting that Ppz phosphatases may regulate ubiquitin metabolism at multiple levels. We also agree with the reviewers that the data suggest Ser57 phosphorylation may be sufficient but not strictly required for many of the phenotypes observed. There are several potential explanations for why this may be the case, which we have attempted to address throughout the manuscript. Specifically, we have provided additional experimentation (Figure 3—figure supplement 2, see more details below) and made extensive revisions throughout the text, including: (i) subsection “Ubiquitin Turnover is Regulated by the Ser57 Position of Ubiquitin”, second paragraph; (ii) subsection “Ser57 phosphomimetic ubiquitin accelerates endocytic trafficking”, last paragraph; (iii) subsection “Ser57 phosphomimetic ubiquitin bypasses an artificial DUB checkpoint”, end of first paragraph; (iv) Discussion, first paragraph. Overall, we believe these new experiments and modifications to the text of the manuscript provide the “more nuanced set of conclusions” requested by reviewers.

The reviewers suggest three approaches to try to address this: a) The stoichiometry of S57 phosphorylation needs to be measured with and without ppz. This can be done using various approaches, but perhaps quantitative proteomics is the best approach. If the stoichiometry is low, you would need to accurately describe the implications of this on your conclusions. It may be the case that normally, a very small fraction of UB is phosphorylated on this site and that pool is more rapidly degraded during the execution of endocytosis.

This is an excellent suggestion – and to address this rigorously we have included absolute quantitative analysis by mass spectrometry. By spiking isotopically labelled heavy standard peptides into affinity purified ubiquitin, we were able to measure absolute quantities of ubiquitin. However, the abundance of Ser57 phosphorylated ubiquitin was below the limit of reliable quantification (even when we load sample to instrument capacity) – as illustrated in Figure 3—figure supplement 2. Although we cannot derive an accurate stoichiometry for Ser57 phosphorylation in biological samples, we can conclude that the abundance is below 0.05% of total ubiquitin in the samples measured. As suggested by reviewers, this extremely low stoichiometry may be consistent with localized regulation of specific pools of ubiquitin. However, it also limits our ability to draw physiological conclusions from expression of the phosphomimic, as it is unlikely that Ser57 phosphoubiquitin is ever present systemically. As requested by reviewers, we have tried to accurately describe the implications of these results throughout the text, including: (i) subsection “Ubiquitin Turnover is Regulated by the Ser57 Position of Ubiquitin”, second paragraph; (ii) subsection “Ser57 phosphomimetic ubiquitin accelerates endocytic trafficking”, last paragraph; (iii) subsection “Ser57 phosphomimetic ubiquitin bypasses an artificial DUB checkpoint”, end of first paragraph; (iv) Discussion, first paragraph. Overall, we believe these new experiments and modifications to the text of the manuscript provide the “more nuanced set of conclusions” requested by reviewers.

b) Further analysis of the ppz1/2 deletion strain which has elevated S57 phosphorylated ubiquitin would strengthen the paper, as this seems a more natural and relevant condition than overexpression of the S57D mutant ubiquitin. For example, what does Mup1 trafficking look like in the ppz1/2 deletion strain (Figure 4), or does the ppz1/2 strain display the same bypass of Hse1-UL36 for Mup1-GFP trafficking (Figure 5) as the S57D mutant ubiquitin strain?

This is an excellent point – and one that we have been working on. However, the results of this line of investigation have led to a parallel manuscript which we are currently preparing for submission.

c) Depending on the results of points 1 and 2, you would need to be able to provide a more nuanced description of the results, as there is concern that some of the conclusions are too strong. In particular, the fact that much of the UB is degraded in the context of the S57D mutant is likely not to be physiological, since the stoichiometry is likely low compared to the apparent case of the phosphomimetic. So a much more nuanced set of conclusions should be made, depending on the outcome.

As described above, we have confirmed the low stoichiometry of Ser57 phosphorylation of ubiquitin. Additionally, we have incorporated multiple revisions throughout the text (described above) to effectively incorporate more nuanced and limited conclusions with respect to results obtained using the S57D mutant. Thus, we feel the revised manuscript effectively conveys the above points raised by reviewers.

d) There is also an issue with the possibility that the phosphomimetic doesn't actually phenocopy the phosphoform. In the case of pS65 UB, it is clear from the literature that the phosphomimetic is not really a phosphomimetic biochemically, as pS65 UB can dramatically activate parkin in vitro but S65D UB has essentially no activity in such assays. Some discussion of this issue or qualifications in the text is likely needed.

We agree with the reviewers on this point. We have clarified this in the text and emphasized how this limits our ability to draw conclusions from the analysis of the Ser57Asp phosphomimetic (subsection “Phosphorylation as an Emerging Mechanism of Ubiquitin Regulation”, first paragraph).

2) Overall, the treatment of the mass spectrometry experiments is insufficiently described. It is unclear for example how error bars are being generated from a SILAC experiment apparently performed only once unless the error bars are coming from different peptides, but this can't really be done for single peptides by SILAC. A detailed description of how the statistics was done is needed.

We have provided additional details on the mass spectrometry analysis in the Materials and methods section of this paper. Additionally, we have removed the previous Figure 1—figure supplement 4 – which contained error bars that reflected averaging the H:L ratio across multiple peptides for the same protein. However, since those data did not contribute substantially to the overall scope or conclusions of this paper (and in fact those data appear more pertinent to our parallel trafficking story) this figure supplement has been removed.

3) There was some concern and discussion concerning the fact that the strains have overexpressed UB in some cases, although the reviewers understand the difficulties in making mutations in the endogenous loci. It might be useful to have blots with all the UB chains collapsed down to mono UB with USP2 in order to know what the actual UB levels are.

This is an excellent point and we have addressed this by performing slot blot analysis of total cell lysates. Briefly, we used yeast strains expressing FLAG-tagged ubiquitin from endogenous loci (*RPL40B* and *RPS31*, both with native promoters and terminators) so we are confident all signal on the slot blot is derived from ubiquitin (Figure 1 and Figure 1—figure supplement 3). (In contrast, blotting with ubiquitin antibodies is likely to yield background signal which would be impossible to measure/control.) This experimental analysis allows us to quantify ubiquitin abundance in cell lysates without having to overexpress ubiquitin.

4) Quite a number of different yeast strains with different ways of expressing UB are used. Perhaps this should be explained.

We agree that the use of different strains can be confusing – and we appreciate the reviewers’ understanding. To address this, we have added a general strain usage statement at the beginning of the “Yeast Strains and Growth Conditions” section of the Experimental Procedures. Additionally, we have re-organized some figures to help make strain usage more consistent. For example, in Figure 1 all ubiquitin levels experiments shown in the main figure correspond to the SEY6210 background, while corresponding analysis in SUB280 has been placed in Figure 1—figure supplement 4.